# Does Generation Require Memorization?
# Creative Diffusion Models using Ambient Diffusion

**Kulin Shah** [1]  **Alkis Kalavasis** [2]  **Adam Klivans** [1]  **Giannis Daras** [3]

## Abstract

There is strong empirical evidence that the state-of-the-art diffusion modeling paradigm leads to models that memorize the training set, especially when the training set is small. Prior methods to mitigate the memorization problem often lead to decrease in image quality. Is it possible to obtain *strong* and *creative* generative models, i.e., models that achieve high generation quality and low memorization? Despite the current pessimistic landscape of results, we make significant progress in pushing the trade-off between fidelity and memorization. We first provide theoretical evidence that memorization in diffusion models is only necessary for denoising problems at low noise scales (usually used in generating high-frequency details). Using this theoretical insight, we propose a simple, principled method to train the diffusion models using noisy data at large noise scales. We show that our method significantly reduces memorization without decreasing the image quality, for both text-conditional and unconditional models and for a variety of data availability settings.

## 1. Introduction

Diffusion models (Song & Ermon, 2019; Ho et al., 2020; Song et al., 2020) have become a widely used framework for unconditional and text-conditional image generation. However, recent works (Carlini et al., 2023; Daras et al., 2023c; 2024b; Somepalli et al., 2022; 2023; Ross et al., 2024) have shown that the trained models memorize the training data and often replicate them at generation time. This issue has raised important privacy and ethical concerns (Somepalli et al., 2022; Tramèr et al., 2022; Appel

et al., 2023), especially in applications where the training set contains sensitive or copyrighted information (Chambon et al., 2022b;a). Carlini et al. (2023) conjectures that the improved performance over alternative frameworks may come *from* the increased memorization (Lee et al., 2024). This raises the following question:

> *Can we improve the memorization of diffusion models without decreasing the image generation quality?*

Prior work has shown that the optimal solution to the diffusion objective is a model that merely replicates the training points (Scarvelis et al., 2023; Biroli et al., 2024; De Bortoli, 2022; Kamb & Ganguli, 2024; Benton et al., 2024). Several papers have tried to understand the reason behind memorization and transition from memorization to generalization (Kadkhodaie et al., 2023; Kamb & Ganguli, 2024). For example, Kamb & Ganguli (2024) showed that the creativity in diffusion modeling happens when the models fail to perfectly minimize their training loss. As the training dataset becomes smaller, overfitting becomes easier, memorization increases and output diversity decreases (Kadkhodaie et al., 2023; Somepalli et al., 2023; Daras et al., 2023c; Gu et al., 2023). Text-conditioning is also known to exacerbate memorization (Carlini et al., 2023; Somepalli et al., 2023; 2022) and text-conditional diffusion models are known to memorize individual training points even when trained on billions of image-text pairs (Carlini et al., 2023; Daras et al., 2023a).

**Related work.** Several methods have been proposed to reduce the memorization in diffusion models (Somepalli et al., 2023; Daras et al., 2023c; Wen et al., 2024; Daras et al., 2024a; Kazdan et al., 2024; Chen et al., 2024). A line of work proposes sampling adaptations that guide the generation process away from training points (Kazdan et al., 2024; Wen et al., 2024; Chen et al., 2024). Wang et al. (2025); Kulikov et al. (2023) propose decreasing the receptive field of the generative model to avoid memorization. Another line of work corrupts the images (Daras et al., 2023c; 2024b) or the text-embedding in text-conditioned image models (Somepalli et al., 2023). Another way of tackling memorization is by using privacy based methods (e.g., differential privacy based methods (Dockhorn et al., 2022; Lin et al., 2023; Wang et al., 2024a) ). These methods, while effective in reducing memorization and improving the privacy, often

*Equal contribution  [1]University of Texas at Austin  [2]Yale University  [3]Massachusetts Institute of Technology. Correspondence to: Kulin Shah <kulinshah@utexas.edu>.

*Proceedings of the 42nd International Conference on Machine Learning*, Vancouver, Canada. PMLR 267, 2025. Copyright 2025 by the author(s).

| **Training image** | **(Wen et al., 2024)** | **(Wen et al., 2024) + Ours** |
|---|---|---|

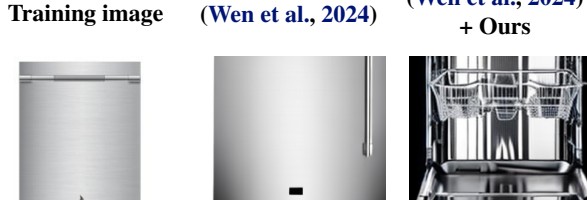

"RISE 24" TriFecta Dishwasher

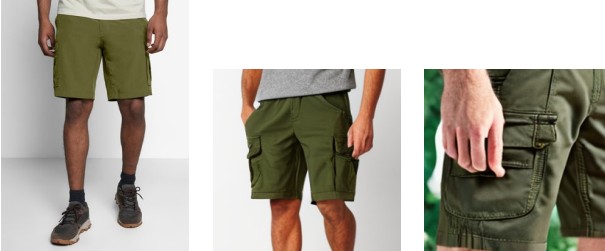

CANYON CARGO - Outdoor shorts - dark moss

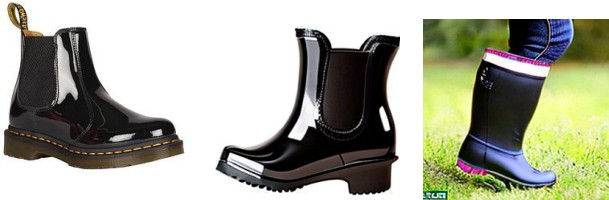

Western Chief Down Hill Trot (Black) Women's Rain Boots

Figure 1. Qualitative results for reducing the memorization of Stable Diffusion 2. Combining our method with (Wen et al., 2024) helps generate novel samples for the above prompts. See Section 3 for our method and Section 5.2 for more details on the experiment.

decrease the image generation quality. Feldman (2020) theoretically showed strong trade-offs between memorization and generalization by showing that memorization is *necessary* for (optimal) *classification*. This raises the natural question of whether this trade-off also applies to *generative modeling*.

The need for memorization in (Feldman, 2020) is associated with the frequencies of different subpopulations (e.g., cats, dogs, etc.) that appear in the dataset. The key observation is that the distribution of the frequencies is usually *heavy-tailed* (Zhu et al., 2014), i.e., roughly speaking in a dataset of size $n$, there will be many classes with frequency around $1/n$. This means that the training algorithm will only observe a single representative from those subpopulations and cannot distinguish between the following two cases: **Case 1.** If the unique example comes from an extremely rare subpopulation (with frequency $\ll 1/n$), then memorizing it has no significant benefits, **Case 2.** If the unique example comes from a subpopulation with $1/n$ frequency, then memorizing it will probably improve the accuracy on the

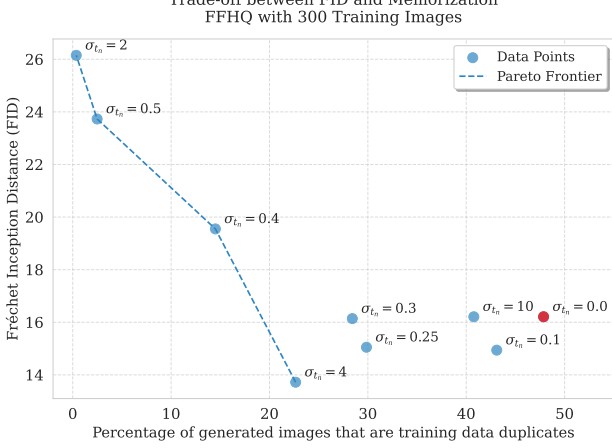

Figure 2. (FID, Memorization) pairs for different values of $\sigma_{t_n}$ used in our proposed Algorithm 1 (presented in Section 3) for training diffusion models from limited data. The standard DDPM objective corresponds to $\sigma_{t_n} = 0$ and it is not in the Pareto frontier. Setting $\sigma_{t_n}$ too low or too high reverts back to the DDPM behavior. Values for $\sigma_{t_n} \in [0.4, 4]$ strike different balances between memorization and quality of generated images. The models in this Figure are trained on only 300 images from FFHQ.

entire subpopulation and decrease the generalization error by $\Omega(1/n)$. Hence, the optimal classifier should memorize these unique examples to avoid paying Case 2 in the error.

The key assumption above seems to break when noise is added to the images. That is because different subpopulations start to merge and the heavy-tails of the weights' distribution disappear. Interestingly, diffusion models learn the (score of the) distribution at different levels of noise. This indicates that, in principle, it is feasible to avoid memorization in the high-noise regime (without sacrificing too much quality). Despite that, regular diffusion model training, e.g., the DDPM (Ho et al., 2020) objective, results to score functions that have attractors around the training points, even for highly noisy inputs, as shown in Figure 3.

The discussion above suggests that it should be possible to train high-quality diffusion models that do not memorize in the high-noise part of the diffusion. It has been empirically established that this part controls the structural information of the outputs and hence the diversity of the generated distribution (Dieleman, 2024; Li & Chen, 2024). To avoid memorization in the high-noise regime, we propose a simple, principled framework that trains the diffusion model only with noisy data at large noise scales. We give theoretical evidence that the noisy targets used for learning leak much less information about the training set and further they are harder to memorize since they are less compressible.

**Our contributions:**

- We propose a simple framework to train diffusion mod-

**Input images**

$\sigma = 17$ $\qquad\qquad\qquad$ $\sigma = 8$

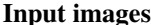

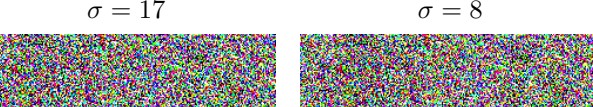

**Outputs of diffusion model trained on 52k images**

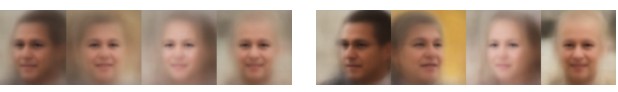

**Outputs of diffusion model trained on 300 images**

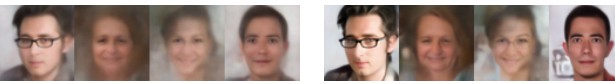

**Outputs of Ambient Diffusion trained on 300 images**

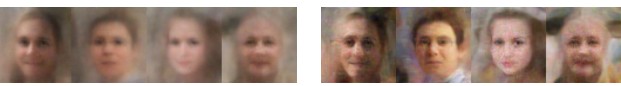

*Figure 3.* Comparison of denoised images under different noise levels and training conditions. Standard diffusion modeling leads to overconfident predictions (row 3) even for very highly noised inputs when it is trained on small datasets. Our algorithm (row 4), has a similar behavior (blurry outputs) to a model trained with significantly more data (row 1), indicating less memorization.

els that achieve reduced memorization and high-quality sample generation even when trained on limited data.

- We experimentally validate our approach on various datasets and data settings, showcasing significantly reduced memorization and improved generation quality compared to natural baselines, in both the unconditional and text-conditional settings.
- On the theory side, we adapt the theoretical framework of (Feldman, 2020) for studying memorization to diffusion models. Based on that, we argue about the necessity of memorizing the training set in different noise scales indicating that memorization is only essential at the low-noise regime.
- We quantify the information leakage of our proposed algorithm in the high-noise regime showing significant benefits over the standard diffusion modeling objective.

## 2. Background and Related Work

### 2.1. Diffusion Modeling

The first step in diffusion modeling is to design a corruption process. For the ease of presentation, we focus on the widely used Variance Preserving (VP) corruption (Ho et al., 2020; Song et al., 2020). We define a sequence of increasing corruption levels indexed by $t \in [0, 1]$, with:

$$X_t = \sqrt{1 - \sigma_t^2} X_0 + \sigma_t Z, \quad Z \sim \mathcal{N}(0, I_d), \quad (1)$$

where the map $\sigma_t := \sigma(t)$ is the noise schedule and $X_0$ is drawn from the clean distribution $p_0$. We remark that our framework extends to other noise schedules, diffusion models (Song & Ermon, 2019; Daras et al., 2023b; Bansal et al., 2022; Karras et al., 2022) and flow matching (Lipman et al., 2022; Liu et al., 2022; Albergo et al., 2023).

Our ultimate goal is to sample from the unknown distribution $p_0$. The key idea behind diffusion modeling is to learn the score functions, defined as $\nabla \log p_t(\cdot)$, for different noise levels $t$, where $X_t \sim p_t$. The latter is related to the optimal denoiser $\mathbb{E}[x_0 | X_t = x_t]$ through Tweedie's formula (Efron, 2011):

$$\nabla \log p_t(x_t) = \frac{\sqrt{1 - \sigma_t^2} \mathbb{E}[x_0 | X_t = x_t] - x_t}{\sigma_t^2}. \quad (2)$$

The conditional expectation is typically learned from the available data with supervised learning over some parametric class of models $\mathcal{H} = \{h_\theta : \theta \in \Theta\}$, using the training objective:

$$J(\theta) = \mathbb{E}_{x_0} \mathbb{E}_{(x_t, t) | x_0} \left[ \| h_\theta(x_t, t) - x_0 \|^2 \right]. \quad (3)$$

Post training, the score function $\nabla \log p_t(x_t)$ is approximated by plugging the optimal solution of (3) to (2). Alternatively, one can train directly for the score function using the noise prediction loss (Ho et al., 2020; Vincent, 2011):

$$J(\theta) = \mathbb{E}_{x_0, x_t, t} \left[ \left\| s_\theta(x_t, t) - \frac{\sqrt{1 - \sigma_t^2} x_0 - x_t}{\sigma_t^2} \right\|^2 \right]. \quad (4)$$

Given access to the score function for different times $t$, one can sample from the distribution of $p_0$ by running the process (Song et al., 2020):

$$dx = \left( -x - \frac{(d\sigma_t/dt)\sigma_t}{1 + \sigma_t^2} \nabla \log p_t(x_t) \right) dt. \quad (5)$$

### 2.2. Memorization in Diffusion Models

The first expectation of (3) is taken over the distribution of $x_0$. The underlying distribution of $x_0$ is continuous, but in practice we only optimize this objective over a finite distribution of training points. Prior work has shown that when the expectation is taken over an empirical distribution $\widehat{p}_0$, the optimal score can be written in closed form (Scarvelis et al., 2023; Biroli et al., 2024; De Bortoli, 2022; Kamb & Ganguli, 2024; Benton et al., 2024). Specifically, the optimal score for the empirical distribution, which corresponds to a finite amount of examples $S$, can be written as:

$$\widehat{s}_*(x_t, t) = \frac{1}{\sigma_t^2} \frac{1}{\sum_{x_0 \in S} \mathcal{N}(x_t; \sqrt{1 - \sigma_t^2} x_0, \sigma_t I)} \cdot$$

$$\cdot \sum_{x_0 \in S} \left( \sqrt{1 - \sigma_t^2} x_0 - x_t \right) \mathcal{N}(x_t; \sqrt{1 - \sigma_t^2} x_0, \sigma_t I).$$

Intuitively, each point $x_0$ in the finite sample $S$ (i.e., the empirical distribution $\widehat{p}_0$) is pulling the noisy iterate $x_t$ towards itself, where the weight of the pull depends on the distance of each training point to the noisy point. The above solution will lead to a diffusion model that only *replicates* the training points during sampling (Scarvelis et al., 2023; Kamb & Ganguli, 2024). Hence, any potential creativity that is observed experimentally in diffusion models comes from the failure to perfectly optimize the training objective.

### 2.3. Ambient Score Matching

One way to mitigate memorization is to never see the training data. Recent techniques for training with corrupted data allow learning of the score function without ever seeing a clean image (Daras et al., 2024b; 2023a; Kawar et al., 2023; Bai et al., 2024; Wang et al., 2024b; Rozet et al., 2024). Consider the case where we are given samples from a noisy distribution $p_{t_n}$ (where $t_n$ stands for $t$-nature) and we desire to learn the score at time $t$ for $t > t_n$. The Ambient Score Matching loss (Daras et al., 2024b), defined as:

$$J_{\text{ambient}}(\theta) = \mathbb{E}_{x_{t_n}} \mathbb{E}_{(x_t, t)|x_{t_n}} \left[ \left\| \frac{\sigma_t^2 - \sigma_{t_n}^2}{\sigma_t^2 \sqrt{1 - \sigma_{t_n}^2}} h_\theta(x_t, t) \right. \right.$$

$$\left. \left. + \frac{\sigma_{t_n}^2}{\sigma_t^2} \sqrt{\frac{1 - \sigma_t^2}{1 - \sigma_{t_n}^2}} x_t - x_{t_n} \right\|^2 \right], \quad (6)$$

can learn the conditional expectation $\mathbb{E}[x_0|x_t]$ (similar to Equation (3)) without ever looking at clean data from $p_0$. The intuition behind this objective is that to denoise the noisy sample $x_t$, we need to find the direction of the noise and then rescale it appropriately. The former can be found by denoising to an intermediate level $t_n$ and the rescaling ensures that we denoise all the way to the level of clean images. Once the conditional expectation $\mathbb{E}[x_0|x_t]$ is recovered, we get the score by using Tweedie's Formula.

We remark that this objective can only be used for $t > t_n$. While there are ways to train for $t \leq t_n$ without any clean data (e.g. see (Daras et al., 2023a; 2024b; Bai et al., 2024; Wang et al., 2024b; Rozet et al., 2024)), this leads to performance deterioration unless a massive noisy dataset is available (Daras et al., 2024a). For what follows, we refer to Eq.(3) as the DDPM training objective and to Eq.(6) as the Ambient Diffusion training objective for noisy data.

## 3. Method

We are now ready to present our framework for training diffusion models with limited data that will allow creativity without sacrificing quality. Our key observation is that the diversity of the generated images is controlled in the high-noise part of the diffusion trajectory (Dieleman, 2024; Li &

Chen, 2024). Hence, if we can avoid memorization in this regime, it is highly unlikely that we will replicate training examples at inference time, even if we memorize at the low-noise part. Our training algorithm can "copy" details from the training samples and still produce diverse outputs.

Our training framework is presented in Algorithm 1. It works by splitting the diffusion training time into two parts, $t \leq t_n$ and $t > t_n$, where $t_n{}^1$ ($t$-nature) is a free parameter to be controlled. For the regime, $t \leq t_n$, we train with the regular diffusion training objective, and (assuming perfect optimization) we know the exact score, which is as given in Section 2.2. To train for $t > t_n$, we first create the set $S_{t_n}$ which has *one* noisy version of each image in the training set. Then, we train using the set $S_{t_n}$ and the Ambient Score Matching loss introduced in Section 2.3.

It is useful to build some intuition about why this algorithm avoids memorization and at the same time produces high-quality outputs. Regarding memorization: 1) the learned score function for times $t \geq t_n$ does not point directly towards the training points since Ambient Diffusion aims to predict the noisy points (recall that the optimal DDPM solution points towards scalings of the training points) and 2) the noisy versions $x_{t_n}$ are harder to memorize than $x_0$, since noise is not compressible. At the same time, if the dataset size were to grow to infinity, both our algorithm and the standard diffusion objective would find the true solution: the score of the underlying continuous distribution. In fact, Algorithm 1 learns the same score function for times $t \leq t_n$ as DDPM. This contributes to generating samples with high-quality details, copied from the training set.

## 4. Theoretical Results

### 4.1. Information Leakage

In this section, we attempt to formalize the intuition of why our proposed algorithm reduces memorization of the dataset. We start by showing the following Lemma that characterizes the sampling distribution of our algorithm for $t = t_n$.

**Lemma 4.1** (Ambient Diffusion solution at $t_n$). *Let $S_{t_n}$ be the noisy training set as in L1 of Algorithm 1. For a fixed $S_{t_n}$, let $\hat{p}_{t_n}$ be the distribution at time $t = t_n$ that arises by using the score of Algorithm 1 in the reverse process of Eq.(5) initialized at $\mathcal{N}(0, I_d)$. It holds that $\hat{p}_{t_n} = \frac{1}{|S_{t_n}|} \sum_{x_{t_n} \in S_{t_n}} \delta(x - x_{t_n})$.*

For the proof, we refer to Section D.1.1. This Lemma extends the result of Kamb & Ganguli (2024) from the standard diffusion objective of Eq.(3) to the training objective of Eq.(6). Given this result, we can compare the information leakage of Ambient Diffusion at time $t_n$ compared to the

---

[1]We often use the symbol $n$ for sample size; the notation $t_n$ is unrelated to the size $n$.

attraction to $x_0$ | weight of attraction

optimal distribution $\widehat{q}_{t_{\mathrm{n}}}$ learned by DDPM at that time.

**Lemma 4.2** (Information Leakage)**.** *Consider point $x_0 \sim \mathcal{N}(\mu, \Sigma)$, a set $A$ of size $m$ generated i.i.d. by $\widehat{p}_{t_{\mathrm{n}}}$ (optimal ambient solution at time $t_{\mathrm{n}}$ with input $x_0$) and a set $D$ of size $m$ generated i.i.d. by $\widehat{q}_{t_{\mathrm{n}}}$ (optimal DDPM solution at time $t_{\mathrm{n}}$ with input $x_0$). Then the mutual information satisfies:*
$$I(D; x_0) = m \cdot I(A; x_0) = \frac{m}{2} \log \det(\frac{1 - \sigma_{t_{\mathrm{n}}}^2}{\sigma_{t_{\mathrm{n}}}^2} \Sigma + I).$$

For the proof, we refer to Section D.1.2. The above means that DDPM leaks much more information about the training point compared to Ambient Diffusion, when asked to generate a collection of samples from the model at time $t_{\mathrm{n}}$. Another way to see it, is that given $m$ samples from DDPM at time $t_{\mathrm{n}}$, one can get an estimator for $x_0$ with error $\mathrm{poly}(1/m)$, while with Ambient Diffusion, no consistent estimation is possible. As expected, as $\sigma_{t_{\mathrm{n}}} \approx 0$, then no noise is added to create $S_{t_{\mathrm{n}}}$ and hence the mutual information blows up. On the other extreme, as $\sigma_{t_{\mathrm{n}}} \approx 1$, then the models reveal no information about the original point. If the dataset contains multiple points, similar results about the mutual information can be obtained (see Section D.1.3).

The above indicate that Ambient Diffusion can only memorize the noisy images. Our justification for the improved performance in practice is that memorizing noise is much harder since noise is not compressible. Even if the noisy images are perfectly memorized, they do not contain enough information to perfectly recover the training set (as shown above) and hence creativity will emerge. A possible conjecture is that under reasonable smoothness assumptions the concatenation of Ambient Diffusion (i.e., of a non-memorized trajectory (up to $t_{\mathrm{n}}$)) and of DDPM (i.e., of a memorized one (from $t_{\mathrm{n}}$ to 0)) will not lead to memorized outputs. Under this conjecture, controlling the high noise case is all you need to decrease memorization, and this is what our algorithm achieves. Showing some non-trivial lower bound between the distribution learned by our algorithm and the distribution learned by DDPM is a challenging theoretical problem that remains to be addressed.

### 4.2. Connections to Feldman (2020)

In the previous section, we discussed ways to reduce the memorization. In this section, we consider what is the price to pay for reduced memorization, i.e., we analyze the trade-off between memorization and fidelity.

While there is significant amount of empirical research on connections between memorization and generation for diffusion models, our rigorous theoretical understanding is still lacking. In terms of theory, there are many works studying memorization-generalization trade-offs for machine learning algorithms (Feldman, 2020; Feldman & Zhang, 2020; Brown et al., 2021; Attias et al., 2024; Cheng et al., 2022; Livni, 2024; Brown et al., 2022) with several connections

to differential privacy and stability in learning (Feldman, 2020; Russo & Zou, 2019; Xu & Raginsky, 2017; Bousquet & Elisseeff, 2002; Steinke & Zakynthinou, 2020; Bassily et al., 2018). Our work studies this trade-off in diffusion models, inspired by the work of (Feldman, 2020).

**Section Overview.** To facilitate the understanding of this Section, we provide an overview of the results. In Section 4.2.1, we define the distribution to be learned as a mixture of distributions of subpopulations (e.g., dogs, cats, etc.) with unknown mixing weights. This distribution is learned given a finite set $Z$ of size $n$ and we are interested in the generalization error of the trained model (at some fixed noise scale $\sigma_t$). In Theorem 4.3 we express this generalization error into two terms, one of which is the error of the algorithm for populations that are seen only once during training. We consider that the trained model "memorizes" when the error of these rare examples is small. Due to the error decomposition, generalization is related to the memorization error and its multiplying constant $\tau_1$ that appears in Theorem 4.3. In Section 4.2.3 we analyze how this constant changes for different noise levels under the assumption of (Zhu et al., 2014; Feldman, 2020) that the mixing weights are heavy-tailed. We argue that when the noise level is small, $\tau_1$ is large and due to the decomposition, the only way to achieve good generalization is to memorize. For high noise levels, $\tau_1$ becomes smaller and hence it is in principle possible to achieve generalization without excessive memorization.

### 4.2.1. SUBPOPULATIONS MODEL OF FELDMAN (2020)

Let us consider a continuous data domain $X \subseteq \mathbb{R}^d$ (e.g., images). We model the data distribution as a mixture of $N$ fixed distributions $M_1, ..., M_N$, where each component corresponds to a subpopulation (e.g., dogs, cats, etc.). For simplicity, we follow Feldman (2020) and assume that each component $M_i$ has disjoint support $X_i$ (this can be relaxed, see Remark B.3). Without loss of generality, let $X = \cup_i X_i$.

We will now describe the procedure of (Feldman, 2020) that assigns frequencies to each subpopulation of the mixture.

1. Consider a list of frequencies $\pi = (\pi_1, \pi_2, ..., \pi_N)$.
2. For each component $i \in [N]$ of the mixture, select randomly and independently an element $p_i$ from $\pi$.
3. Finally, to obtain the mixing weights, we normalize the elements $p_1, ..., p_N$, i.e., the weight of component $i$ is $D_i = \frac{p_i}{\sum_{j \in [N]} p_j}$.

We denote by $\mathcal{D}_\pi$ the distribution over the mixing coefficients tuple $(D_1, ..., D_N)$. A sample $D \sim \mathcal{D}_\pi$ is just a list of the normalized frequencies of the $N$ subpopulations. If $D \sim \mathcal{D}_\pi$, then we can define the true mixture as

$$M_D(x) = \sum_{i \in [N]} D_i \, M_i(x).$$

The above random distribution corresponds to the subpopulations model introduced by Feldman (2020).

### 4.2.2. ADAPTATION TO DIFFUSION

As explained in the Background Section 2, one way to train a generative model in order to generate from the target $M_D$ is to estimate the score function $\nabla_x \log M_{D_t}$ for all levels of noise indexed by $t$. For the analysis of this Section, we consider the case of a single fixed $t$. We define learning algorithms $A$ as (potentially randomized) mappings from datasets $Z$ to score functions $s_\theta \sim A(Z)$.

As in Feldman (2020), we are interested about the expected error of $A$ conditioned on dataset being equal to $Z \in X^n$ as

$$\overline{\mathrm{err}}(\pi, A|Z) = \mathbb{E}_{D \sim \mathcal{D}_\pi(\cdot|Z)} \mathbb{E}_{s_\theta \sim A(Z)} \mathrm{err}_{M_D}(s_\theta) \,,$$

where $D \sim \mathcal{D}_\pi$ is a (random) collection of mixing weights and $\mathrm{err}_{M_D}(s_\theta) = \mathbb{E}_{x_0 \sim M_D} L(s_\theta; x_0)$ for some loss function $L$ is the expected loss of the score function $s_\theta$ under the true population $M_D$. The results we will present shortly are agnostic to the choice of $L$, but the reader should think of $L$ as the noise prediction loss used in (4) for a fixed time $t$.

The quantity $\overline{\mathrm{err}}(\pi, A|Z)$ measures the generalization error of the score function of the learning algorithm $A$ conditional on the training set being $Z$. We will show that the population loss of an algorithm given a dataset $Z$ is at least:

1. its loss on the *unseen* part of the domain, i.e., the population loss in $X \setminus Z$ plus
2. its loss on the elements of $Z$ that belong to subpopulations that are represented only *once* in $Z$ (i.e., the dataset contains a single image of a dog or a single image of a car). This loss, denoted by $\mathrm{err}_Z(A, 1)$, is scaled up by a coefficient $\tau_1$, which expresses the "likelihood" of having such subpopulations.

Typically, we define:

$$\tau_1 = \frac{\mathbb{E}_{\alpha \sim \overline{\pi}}[\alpha^2 (1-\alpha)^{n-1}]}{\mathbb{E}_{\alpha \sim \overline{\pi}}[\alpha (1-\alpha)^{n-1}]} \,,$$

where $\overline{\pi}$ is the marginal distribution $\overline{\pi}(a) = \Pr_D[D_i = a]$. Note that, because the random process of picking the mixing weights is run independently for any $i \in [N]$, the marginal is the same across different $i$'s (and hence we omit the index $i$ from $\overline{\pi}$). We are now ready to present our result.

**Theorem 4.3** (Informal, see Theorem B.2)**.** *It holds that*

$$\overline{\mathrm{err}}(\pi, A|Z) \geq \overline{\mathrm{err}}_{\mathrm{unseen}}(\pi, A|Z) + \tau_1 \cdot \mathrm{err}_Z(A, 1) \,.$$

The above result can be extended to subpopulations represented by 2 or more examples in $Z$ (see Appendix B). The above inequality relates the population error of the model with its loss on some parts of the training set. The crucial

parameter that relates the two quantities is the coefficient $\tau_1$. If the coefficient $\tau_1$ is large, it means that if the model does not fit the "rare examples" of the dataset, it will have to pay roughly $\tau_1$ in the generalization error. As shown by (Feldman, 2020), $\tau_1$ is controlled by how much heavy-tailed is the distribution of the frequencies of the mixture model. This is the topic of the next section, where we also investigate the effect of adding noise to the training set.

### 4.2.3. HEAVY TAILS AND THE ROLE OF NOISE

In this section, we are going to formally explain what it means for the frequencies of the original dataset to be heavy-tailed (Zhu et al., 2014; Feldman, 2020). This heavy-tailed structure will then allow us to control the generalization error in Theorem 4.3. We will be interested in subpopulations that have only one representative in the training set $Z$ (these are the examples that will cost roughly $\tau_1$ in the error of Theorem 4.3). We will refer to them as *single* subpopulations. For this to happen given that $|Z| = n$, it should be roughly speaking the case where some frequencies $D_i$ are of order $1/n$. The quantity that controls how many of the frequencies $D_i$ will be of order $1/n$ is the mass that the distribution $\overline{\pi}(a) = \Pr_D[D_i = a]$ assigns to the interval $[1/(2n), 1/n]$. Typically, we will call a list of frequencies $\pi$ *heavy-tailed* if

$$\mathrm{weight}\left(\overline{\pi}, \left[\frac{1}{2n}, 1/n\right]\right) = \Omega(1) \,. \tag{7}$$

In words, there should be a constant number of subpopulations with frequencies of order $O(1/n)$. This definition is important because it can then lower bound the value $\tau_1$ in Theorem 4.3 and hence it can lower bound the generalization loss of not fitting single subpopulations.

**Lemma 4.4** (Informal, see Lemma B.4 and Lemma 2.6 in (Feldman, 2020))**.** *Consider a dataset of size $n$ and assume that $\pi$ is heavy-tailed, as in (7). Then $\tau_1 = \Omega(1/n)$.*

On the contrary, when $\pi$ is not heavy-tailed, $\tau_1$ will be small and hence generalization is not hurt by not memorizing (see Lemma B.5). Next, we are going to inspect how the noise scale affects the heavy-tailed structure of the frequencies and hence the value of $\tau_1$. For an illustration, we will consider the most standard model, that of a mixture of Gaussian subpopulations (similar results are expected for more general population models; we note that the previous results can be naturally adapted for the GMM and other cases, see Remark B.3 and the discussion in (Feldman, 2020)). Let us consider a density $q_0 = \sum_{i=1}^N w_i \mathcal{N}(\mu_i, I) = \sum_i w_i \mathcal{N}_i$. We will say that two components $\mathcal{N}_i, \mathcal{N}_j$ are $\epsilon$-separated if $\mathrm{TV}(\mathcal{N}_i, \mathcal{N}_j) > 2\epsilon$ and can be $\epsilon$-merged if $\mathrm{TV}(\mathcal{N}_i, \mathcal{N}_j) \leq \epsilon$. If $\mathcal{N}_i$ and $\mathcal{N}_j$ are merged, we consider that the new coefficient is $w_i$ class $j$

**Lemma 4.5** (Informal, see Section B.4)**.** *Consider the GMM density $q_0$ and let $q_t$ be the density of the forward diffusion*

*Table 1.* FID and Memorization results comparing DDPM and Algorithm 1. Memorization is measured as DINOv2 similarity between generated samples and their nearest training neighbors. We achieve same or better FID with significantly lower memorization.

| | | # Train Images | | | | | |
|---|---|---|---|---|---|---|---|
| | | 300 | | 1k | | 3k | |
| | | DDPM | Ours | DDPM | Ours | DDPM | Ours |
| CIFAR-10 | FID | 25.1 | **23.91** | 10.46 | **10.36** | 14.73 | **14.26** |
| | S>0.9 | 78.96 | **44.84** | 75.86 | **69.08** | 53.40 | **52.24** |
| | S>0.925 | 67.2 | **20.22** | 57.98 | **47.26** | 11.92 | **11.36** |
| | S>0.95 | 56.56 | **9.64** | 43.44 | **26.34** | 0.08 | **0.06** |
| FFHQ | FID | 16.21 | **15.05** | 12.26 | **11.3** | **6.42** | 6.46 |
| | S>0.85 | 63.38 | **49.68** | 55.36 | **32.08** | 21.58 | **20.08** |
| | S>0.875 | 55.48 | **40.01** | 43.82 | **17.48** | 4.98 | **4.53** |
| | S>0.9 | 47.86 | **29.86** | 33.92 | **7.52** | 0.46 | **0.42** |
| ImageNet | FID | —— | | 50.2 | **47.19** | 40.66 | **39.87** |
| | S>0.9 | —— | | 54.72 | **26.68** | 32.86 | **28.40** |
| | S>0.925 | —— | | 41.66 | **15.56** | 12.32 | **9.44** |
| | S>0.95 | —— | | 25.86 | **5.54** | 6.08 | **4.02** |

*process at time $t$ with schedule $\sigma_t \in [0,1]$. Consider any pair of components $\mathcal{N}_i, \mathcal{N}_j$ in $q_0$ with total variation $C_{ij}$ for some absolute constant $C_{ij}$ and let $\mathcal{N}_i^t, \mathcal{N}_j^t$ be the associated distributions in $q_t$.*

- *(Low Noise) If $\sigma_t \leq \sqrt{1 - (2\epsilon/C_{ij})^2}$, then $\mathcal{N}_i^t, \mathcal{N}_j^t$ are $\epsilon$-separated.*
- *(High Noise) If $\sigma_t \geq \sqrt{1 - (\epsilon/C_{ij})^2}$, then $\mathcal{N}_i^t, \mathcal{N}_j^t$ are $\epsilon$-merged with coefficient $w_i + w_j$.*

For a more formal treatment, we refer to Section B.4. The above Lemma has the following interpretations. If the noise level is small, the originally separated subpopulations (at $t = 0$) will remain separated. This implies that if the frequencies (i.e., the mixing weights) were originally heavy-tailed (as in the above discussion), they will remain heavy-tailed even in the low-noise regime, i.e. Lemma 4.4 applies ($\tau_1$ is large). On the other side, as we increase $t$, the clusters start to merge and the heavy-tailed distribution of the mixing coefficients becomes lighter (until all the clusters are merged into a single one). Hence, $\tau_1$ will be small. This conceptually indicates that there is no reason for memorizing the training noisy images $x_t$ (and hence the original images $x_0$ which do not appear during training).

## 5. Experiments

### 5.1. Memorization in Unconditional Models

We start our experimental evaluation by measuring the memorization and performance of unconditional diffusion models in several controlled settings. Specifically, we train models from scratch on CIFAR-10, FFHQ, and (tiny) ImageNet using 300, 1000 and 3000 training samples. For each one of these settings, we compute the Fréchet Incep-

tion Distance (Heusel et al., 2017) (FID) between 50,000 generated samples and 50,000 dataset samples as a measure of quality. Following prior work (Somepalli et al., 2022; 2023; Daras et al., 2023c), we measure memorization by computing the similarity score (i.e., inner product) of each generated sample to its nearest neighbor in the embedding space of DINOv2 (Oquab et al., 2023). For all these experiments, we compare the performance of Algorithm 1 against the regular training of diffusion models (see Eq.(3)).

**Choice of $t_n$.** Our method has a single parameter $t_n$ that needs to be controlled. We argue that there is an interval $(t_{min}, t_{max})$ that contains reasonable choices of $t_n$. Setting $t_n$ too low, i.e., ($t_n \leq t_{min}$), essentially reverts back to the original algorithm that produces memorized images of good quality. But also, setting $t_n$ too high, i.e., $t_n \geq t_{max}$, will also lead to memorization as there is more time in the sampling trajectory (the interval $[0, t_{max}]$), where we use the memorized score. Values in the range $(t_{min}, t_{max})$ achieve low memorization and strike good balances in the quality-memorization trade-off.

**Decreasing memorization without sacrificing quality.** Most of the prior mitigation strategies for memorization often decrease the image generation quality. Here, we ask: how much do we need to memorize to achieve a given image quality? To answer this, we tune the value $t_n$ to train models using Algorithm 1 that match the FID obtained by DDPM, and we measure their memorization levels. To report memorization, we use three thresholds in the similarities of DINOv2 embeddings that semantically correspond to: i) potentially memorized image, ii) (partially) memorized image, and, iii) exact copy of an image in the training set. The thresholds are tuned separately for each dataset to express these semantics. We present analytic results for 300, 1k and 3k training images from CIFAR-10, FFHQ and (tiny)-ImageNet in Table 1[2]. As shown, for the same or better FID, our models achieve significantly lower memorization levels. This leads to the surprising conclusion that models learned by the DDPM loss are not Pareto optimal for small datasets. That said, the benefit from our algorithm in both FID and memorization shrinks as the dataset grows.

**Other points in the Pareto frontier.** So far, our goal was to reduce memorization while keeping FID the same as DDPM. However, by appropriately tuning the value $t_n$, we can achieve other points in the Pareto frontier that achieve varying trade-offs between memorization and quality of generated images. We present these results for a model trained on 300 images from FFHQ in Figure 2. We see that setting $\sigma_{t_n} \in [0.4, 4]$ corresponds to Pareto optimal points, while setting the value of $t_n$ too low or too high brings us

---

[2]For tiny ImageNet, we do not report results in the 300 samples setting since there are 200 different classes and so for some of the classes we do not observe any samples.

*Table 2.* Comparison between DDPM, our Algorithm 1 and results obtained by training with only corrupted data (masking or additive Gaussian noise). As shown, our algorithm achieves low memorization since it uses noisy data in the high-noise regime, but it also achieves low FID (contrary to the algorithms only using corrupted data) as it can copy the high-frequency details from the training samples.

| Metric | # Training Images | | | | | | | | | | | |
| --- | --- | --- | --- | --- | --- | --- | --- | --- | --- | --- | --- | --- |
| | 300 | | | | 1k | | | | 3k | | | |
| | DDPM | Masking | Noise | Ours | DDPM | Masking | Noise | Ours | DDPM | Masking | Noise | Ours |
| FID | 16.21 | 23.40 | 27.92 | **15.05** | 12.26 | 15.73 | 25.57 | **11.3** | **6.42** | 7.44 | 16.28 | 6.46 |
| Sim $> 0.85$ | 63.38 | 53.73 | 29.12 | 49.68 | 55.36 | 38.74 | 14.83 | 32.08 | 21.58 | 19.74 | 12.08 | 20.08 |
| Sim $> 0.875$ | 55.48 | 41.37 | 18.73 | 40.01 | 43.82 | 22.94 | 9.37 | 17.48 | 4.98 | 4.56 | 3.32 | 4.53 |
| Sim $> 0.9$ | 47.86 | 30.34 | 10.60 | 29.86 | 33.92 | 10.08 | 6.49 | 7.52 | 0.46 | 0.43 | 0.36 | 0.42 |

back to the DDPM performance, as expected. For $\sigma_{t_{\mathrm{n}}} = 4$, we almost match the FID that DDPM gets with 1000 images, while we only use 300 images for training, establishing our Algorithm as much more data-efficient than DDPM. We also compute the Frechet distance using Dinov2 and report the result in 3. The result shows that our method improves the quality of generated images.

*Table 3.* FD$_{\mathrm{DinoV2}}$ Scores for Baseline and Our Method

| Method | FD$_{\mathbf{DinoV2}}$ |
| --- | --- |
| Baseline | 353.19 |
| Ours ($\sigma$=0.1) | 347.21 |
| Ours ($\sigma$=0.25) | 344.60 |
| Ours ($\sigma$=0.3) | 358.23 |
| Ours ($\sigma$=0.4) | 371.81 |
| Ours ($\sigma$=0.5) | 382.03 |

**Mean and median similarity of generated samples.** To understand the average similarity of the generated samples to the training samples, we also calculate the mean and median of the similarity score of 50,000 generated samples in the same setting as in Figure 2 (300 training samples from FFHQ). We report our results in Table 4. We see that the mean and median similarity of generated samples is significantly lower for our method.

*Table 4.* Comparison of Mean and Median Similarity Values

| Method | Mean | Median |
| --- | --- | --- |
| DDPM | 0.8826 | 0.8931 |
| Ours ($\sigma = 0.1$) | 0.8854 | 0.8748 |
| Ours ($\sigma = 0.25$) | 0.8518 | 0.8491 |
| Ours ($\sigma = 0.3$) | 0.8473 | 0.8475 |
| Ours ($\sigma = 0.4$) | 0.8287 | 0.8292 |
| Ours ($\sigma = 0.5$) | 0.8060 | 0.8069 |

**Comparison with other mitigation strategies.** For completeness, we include comparisons with two other mitigation

strategies that reduce memorization in the unconditional setting. These methods are known to achieve lower memorization but at the expense of FID. We compare with a model trained on linearly corrupted data (random inpainting), as in the work of (Daras et al., 2023c), and a model trained with only noisy data as in (Daras et al., 2024b). We present the results in Table 2. As shown, our algorithm produces superior behavior as it achieves lower memorization for the same or better FID. The superior performance comes from the ability our method has to generate high-frequency details, contrary to the existing methods that only use solely noisy data and are not capable of such behavior.

### 5.2. Memorization in Text-Conditional Models

We continue our evaluation in text-conditional models. Here, the primal source of memorization is the text-conditioning itself. Wen et al. (2024) observe that for certain trigger prompts, the prediction of the network always converges to the same training point, independent of the image initialization. Our method mitigates image memorization by training with noisy images, so by itself, it cannot mitigate memorization that arises from the text-conditioning. However, we will show that when we combine our method with strategies that mitigate the impact of text memorization, we achieve state-of-the-art results in memorization reduction while keeping the quality of the generated images high.

Following prior work (Somepalli et al., 2023), we finetune Stable Diffusion on 10k image-text pairs from a curated subset of LAION (Schuhmann et al., 2022) and we measure image quality and memorization of the resulting models. We compare with existing state-of-the-art methods for reducing memorizing arising from the text-conditioning. Specifically, we compare with the work of Somepalli et al. (2023) where corruption is added to the text-embedding during training and with the work of Wen et al. (2024) where the model is explicitly trained to pay attention to the visual content (for details, we refer the reader to the associated papers).

We include all the results in Table 5. As shown, the combination of our work with existing methods achieves state-of-

*Table 5.* Memorization and FID results for text-conditional models. Sim denotes the average similarity between a generated sample and its nearest neighbor in the dataset, while 95% is the 95% percentile of the similarities distribution. CLIP measures the image-text alignment. The combination of our method with existing methods from Somepalli et al. (2023) (S23) and Wen et al. (2024) (W24) achieves strong CLIP/FID results with reduced memorization.

| Method | Sim | 95% | CLIP | FID |
|---|---|---|---|---|
| *Without text mitigation:* | | | | |
| Baseline | 0.378 | 0.649 | **0.306** | **18.18** |
| Ours | **0.373** | **0.636** | 0.305 | 18.34 |
| *Text mitigation:* | | | | |
| S23 | 0.319 | 0.573 | 0.302 | **20.55** |
| S23+ ours | 0.308 | 0.547 | **0.306** | 21.30 |
| W24 | 0.208 | 0.300 | 0.293 | 21.44 |
| W24+ ours | **0.192** | **0.267** | 0.293 | 20.74 |

the-art memorization performance while performing on par in terms of image quality. As expected, without any text-mitigation our algorithm fails to improve significantly the memorization since the model remains heavily reliant on the text-conditioning, effectively ignoring the visual content.

## 6. Limitations, Conclusion and Future Work

Our work provides a positive note on the rather pessimistic landscape of results regarding the memorization-quality trade-off in diffusion models. We manage to push the Pareto frontier in various data availability settings for both text-conditional and unconditional models. We further provide theoretical evidence for the plausibility of generation of diverse structures without memorization. We remark that our method does not come with any privacy guarantees or optimality properties and that despite some encouraging first theoretical evidence an end-to-end analysis for the proposed algorithm is currently lacking. We believe that these constitute exciting research directions for future research.

## Acknowledgements

Kulin Shah and Adam Klivans are supported by the NSF AI Institute for Foundations of Machine Learning (IFML). Alkis Kalavasis is supported by the Institute for Foundations of Data Science at Yale (FDS). Giannis Daras is supported by the NSF AI Institute for Foundations of Machine Learning (IFML) and the Computer Science Artificial Intelligence Laboratory at MIT (CSAIL).

## Impact Statement

This paper presents work whose goal is to advance the field of Machine Learning. Machine learning models that memorize training data pose significant privacy and security risks, especially when trained on sensitive or copyrighted datasets. Our methods improve the existing memorization-generation trade-offs. Hence, this paper contributes to safer and more ethical deployment of ML systems.

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

## A. Our Algorithm

In this section, we present the pseudo-code of our algorithmic approach, which combines DDPM and Ambient Diffusion. Details about the approach appear in Section 3.

---

**Algorithm 1** Algorithm for training diffusion models using limited data.

---

**Require:** untrained network $h_\theta$, set of samples $S$, noise level $t_\mathrm{n}$, noise scheduling $\sigma(t)$, batch size $B$, diffusion time $T$

1: $S_{t_\mathrm{n}} \leftarrow \{\sqrt{1 - \sigma_{t_\mathrm{n}}^2} x_0^{(i)} + \sigma_{t_\mathrm{n}} \epsilon^{(i)} | x_0^{(i)} \in S, \epsilon^{(i)} \sim \mathcal{N}(0, I_d)\}$     ▷ Noise the training examples at a specified level.

2: **while** not converged **do**

3:   Form a batch $\mathcal{B}$ of size $B$ uniformly sampled from $S \cup S_{t_\mathrm{n}}$

4:   loss $\leftarrow 0$                          ▷ Initialize loss.

5:   **for** each sample $x \in \mathcal{B}$ **do**

6:    $\epsilon \sim \mathcal{N}(0, I)$                       ▷ Sample noise.

7:    **if** $x \in S_{t_\mathrm{n}}$ **then**

8:     $x_{t_\mathrm{n}} \leftarrow x$                  ▷ We are dealing with a noisy sample.

9:     $t \sim \mathcal{U}(t_\mathrm{n}, T)$            ▷ Sample diffusion time for noisy sample.

10:     $x_t \leftarrow \sqrt{\frac{1 - \sigma_t^2}{1 - \sigma_{t_\mathrm{n}}^2}} x_{t_\mathrm{n}} + \sqrt{\frac{\sigma_t^2 - \sigma_{t_\mathrm{n}}^2}{1 - \sigma_{t_\mathrm{n}}^2}} \epsilon$         ▷ Add additional noise.

11:     loss $\leftarrow$ loss $+ \left\| \frac{\sigma_t^2 - \sigma_{t_\mathrm{n}}^2}{\sigma_t^2 \sqrt{1 - \sigma_{t_\mathrm{n}}^2}} h_\theta(x_t, t) + \frac{\sigma_{t_\mathrm{n}}^2}{\sigma_t^2} \sqrt{\frac{1 - \sigma_t^2}{1 - \sigma_{t_\mathrm{n}}^2}} x_t - x_{t_\mathrm{n}} \right\|^2$    ▷ Ambient Score Matching loss.

12:    **else**

13:     $x_0 \leftarrow x$                 ▷ We are dealing with a clean sample.

14:     $t \sim \mathcal{U}(0, t_\mathrm{n})$            ▷ Sample diffusion time for clean sample.

15:     $x_t \leftarrow \sqrt{1 - \sigma_t^2} x_0 + \sigma_t \epsilon$             ▷ Add noise.

16:     loss $\leftarrow$ loss $+ \| h_\theta(x_t, t) - x_0 \|^2$     ▷ Regular Denoising Score Matching loss.

17:    **end if**

18:   **end for**

19:   loss $\leftarrow \frac{\text{loss}}{B}$                    ▷ Compute average loss.

20:   $\theta \leftarrow \theta - \eta \nabla_\theta \text{loss}$       ▷ Update network parameters via backpropagation.

21: **end while**

---

## B. Subpopulations Model and Connections to Diffusion Models

In this section, we present a more extensive exposition of the framework of the work of (Feldman, 2020). Moreover, we adapt this framework to diffusion models.

### B.1. Subpopulations Model of Feldman (2020)

Let us recall the subpopulations model of (Feldman, 2020). Let us consider a continuous data domain $X \subseteq \mathbb{R}^d$. We model the data distribution as a mixture of $N$ fixed distributions $M_1, ..., M_N$, where each component corresponds to a subpopulation. For simplicity, we follow Feldman (2020) and assume that each component $M_i$ has disjoint support $X_i$ (we can relax this condition, see Remark B.3). Without loss of generality, let $X = \cup_i X_i$. We will now describe the procedure of (Feldman, 2020) that assigns frequencies to each subpopulation of the mixture.

1. First consider a (fixed) list of frequencies $\pi = (\pi_1, \pi_2, ..., \pi_N)$.

2. For each component $i \in [N]$ of the mixture, we select randomly and independently an element $p_i$ from the list $\pi$.

3. Finally, to obtain the mixing weights, we normalize the weights $p_1, ..., p_N$, i.e., the weight of component $i$ is $D_i = \frac{p_i}{\sum_{j \in [N]} p_j}$.

We summarize the above as follows:

**Definition B.1** (Random Frequencies (Feldman, 2020))**.** For the mixing weights, we first consider a list of subpopulation frequencies $\pi = (\pi_1, ..., \pi_N)$. The procedure is the following: we randomly pick $p_i$ from the list $\pi$ for any index $i \in [N]$ and then we normalize ($p_i / \sum_j p_j$ denotes the frequency of subpopulation $i$). We denote by $\mathcal{D}_\pi$ the distribution over probability mass functions on $[N]$ induced by the above procedure.

A sample $D \sim \mathcal{D}_\pi$ is just a list of the frequencies of the $N$ subpopulations.

We also denote by $\bar{\pi}$ the resulting marginal distribution over the frequency of any single element in $i$, i.e.,

$$\bar{\pi}(a) = \Pr_{D \sim \mathcal{D}_\pi} [D_i = a] . \tag{8}$$

Hence, if $D \sim \mathcal{D}_\pi$, then we can define the true mixture as

$$M_D(x) = \sum_{i \in [N]} D_i M_i(x) .$$

The above random distribution corresponds to the subpopulations model introduced by Feldman (2020). Intuitively the choice of the random coefficients for the mixture corresponds to the fact that the learner does not know the true frequencies of the subpopulations.

## B.2. Adaptation of (Feldman, 2020)'s Result to Diffusion Models

As explained in the Background Section 2, one way to train a generative model is to estimate the score function $\nabla \log M_{D_t}$ for all levels of noise indexed by $t$. For the analysis of this Section, we consider the case of a single fixed $t$. We define learning algorithms $A$ as (potentially randomized) mappings from datasets $Z$ to *score functions* $s_\theta \sim A(Z)$. We further define the expected error of $A$ conditioned on dataset being equal to $Z \in X^n$ (eventually $Z$ will be drawn i.i.d. from $M_D$) as

$$\overline{\mathrm{err}}(\pi, A|Z) = \mathbb{E}_{D \sim \mathcal{D}_\pi(\cdot|Z)} \mathbb{E}_{s_\theta \sim A(Z)} \mathrm{err}_{M_D}(s_\theta) ,$$

where $D \sim \mathcal{D}_\pi$ is a random list of frequencies according to Definition B.1 and $\mathrm{err}_{M_D}(s_\theta) = \mathbb{E}_{x_0 \sim M_D} L(s_\theta; x_0)$ for some loss function $L$. The results we will present shortly are agnostic to the choice of loss function $L$, but the reader should think of $L$ as the noise prediction loss used in (4) for a fixed time $t$.

We remark that the quantity $\overline{\mathrm{err}}(\pi, A|Z)$ measures the generalization error of the output score function (according to loss function $L$) of the learning algorithm $A$ conditional on the training set being $Z$. We will relate this generalization error with the loss in the training set. Recall that any subpopulation $i \in [N]$ of the mixture is associated with a domain $X_i$ (and $X_i \cap X_j = \emptyset$ for $i \neq j$).

Let $n$ be the training set size. For any $\ell \in [n]$, consider all the subpopulations $I_\ell \subseteq [N]$ such that $X_i \cap Z = \ell$ for $i \in I_\ell$; in words, $i \in I_\ell$ if there are exactly $\ell$ representatives of cluster $i$ in the dataset $Z$. We can now define $Z_\ell = \{x \in X_i \cap Z : i \in I_\ell\} \subseteq Z$. Note that the sets $Z_1, ..., Z_N$ partition the training set $Z$. For $\ell \in [n]$, we define

$$\mathrm{errn}_Z(A, \ell) = \mathbb{E}_{s_\theta \sim A(Z)} \sum_{x \in Z_\ell} M_{i_x}(x) L(s_\theta; x) . \tag{9}$$

Here $i_x \in [N]$ is the unique index of the component whose support contains $x$. In words, $\mathrm{errn}_Z(A, \ell)$ is the loss of the algorithm $A$ evaluated on the elements of the training set $Z$ that belong to subpopulations will exactly $\ell$ representatives in $Z$.

We show the following result, which is an adaptation of a result of (Feldman, 2020) and relates the population loss with the empirical losses $\mathrm{errn}_Z(A, 1), ..., \mathrm{errn}_Z(A, n)$.

**Theorem B.2.** *Fix a number of samples $n$. Let $\{M_i\}_{i \in [N]}$ be densities of subpopulations over disjoint subdomains $\{X_i\}_{i \in [N]}$. Let $\pi$ be the fixed list of frequencies as in Definition B.1 and let $\bar{\pi}^N$ the marginal distribution of (8). For any learning algorithm $A$ and any fixed dataset $Z \in X^n$, it holds that*

$$\overline{\mathrm{err}}(\pi, A|Z) = \overline{\mathrm{err}}_{\mathrm{unseen}}(\pi, A|Z) + \sum_{\ell \in [n]} \tau_\ell \cdot \mathrm{errn}_Z(A, \ell) , \tag{10}$$

*where*

1. $\overline{\text{err}}_{\text{unseen}}(\pi, A|Z)$ corresponds to the expected $Z$-conditional loss of the algorithm $A$ on the points that do not appear in the training set $Z$.

2. $\tau_\ell$ is a coefficient that corresponds to the weight of having subpopulations with exactly $\ell$ representatives. Given $\mathcal{D}_\pi$ and $\ell \in [n]$, we define

$$\tau_\ell = \frac{\mathbb{E}_{\alpha \sim \overline{\pi}}[\alpha^{\ell+1}(1-\alpha)^{n-\ell}]}{\mathbb{E}_{\alpha \sim \overline{\pi}}[\alpha^{\ell}(1-\alpha)^{n-\ell}]} \, .$$

For the proof we refer to Section D.2.1. The optimal algorithm $A^*$ (without any bound on the complexity of the score function class) is the one that minimizes the RHS in Theorem B.2, which means that

- for any $\ell$, $\text{errn}_Z(A^*, \ell) = 0$ and

- $\overline{\text{err}}_{\text{unseen}}(\pi, A^*|Z)$ is as small as possible.

The above general form relates the population error of the model with its loss on the training set. The crucial parameters that relate the two quantities are the coefficients $\tau_1, ..., \tau_n$. If the coefficient $\tau_1$ is large, it means that if the model does not fit the training examples that appear once in the dataset ("rare examples"), it will have to pay roughly $\tau_1$ in the generalization error. As shown by (Feldman, 2020), $\tau_1$ is controlled by how much heavy-tailed is the distribution of the frequencies of the mixture model. This is the topic of the next section, where we also investigate the effect of adding noise to the training set.

*Remark* B.3 (Gaussian Mixture Models). Subpopulations are often modeled as Gaussians. If the probability of the overlap between the subpopulations is sufficiently small (the means are far), then one can reduce this case to the disjoint one by modifying the components $M_i$ to have disjoint supports while changing the marginal distribution over $Z$ by at most $\delta$ in the TV distance.

## B.3. Heavy-Tailed Distributions of Frequencies

In this section, we are going to formally explain what it means for the frequencies of the original dataset to be heavy-tailed (Zhu et al., 2014; Feldman, 2020). This heavy-tailed structure will then allow us to control the generalization error in Theorem 4.3. Following Feldman (2020), we will assume that the mixing coefficients $D_1, ..., D_N$ are drawn from a heavy-tailed distribution since this is the case in most datasets (Feldman, 2020; Feldman & Zhang, 2020). We will be interested in subpopulations that have only one representative in the training set $Z$ (these are the examples that will cost roughly $\tau_1$ in the error of Theorem 4.3). We will refer to them as *single* subpopulations. For this to happen given that $|Z| = n$, it should be roughly speaking the case where some frequencies $D_i$ are of order $1/n$.

The quantity that controls how many of the frequencies $D_i$ will be of order $1/n$ is the marginal distribution $\overline{\pi}(a) = \Pr_D[D_i = a]$. We first note that the expected number of singleton examples is determined by the weight of the entire tail of frequencies below $1/n$ in $\overline{\pi}$. In particular, one can show (see (Feldman, 2020)) that the expected number of singleton points is at least

$$\frac{n}{2} \cdot \text{weight}(\overline{\pi}, [0, 1/n]), \quad \text{where}$$

$$\text{weight}(\overline{\pi}, [0, 1/n]) := \mathbb{E}_{D \sim \mathcal{D}}\left[\sum_{i \in [N]} D_i \mathbf{1}\{D_i \in [0, 1/n]\}\right] = N \cdot \mathbb{E}_{a \sim \overline{\pi}}[a \mathbf{1}\{a \in [0, 1/n]\}] \, .$$

The above weight function essentially controls how heavy-tailed our distribution over frequencies is. Typically, we will call a list of frequencies $\pi$ heavy-tailed if

$$\text{weight}\left(\overline{\pi}, \left[\frac{1}{2n}, 1/n\right]\right) = \Omega(1) \, .$$

In words, there should be a constant number of subpopulations with frequencies of order $1/n$. This definition is important because it can then lower bound the value $\tau_1$ in Theorem 4.3 and hence it can lower bound the generalization loss of not fitting single subpopulations.

**Lemma B.4** (Lemma 2.6 in (Feldman, 2020)). *For any $\pi$, it holds that $\tau_1 \geq \frac{1}{5n} \cdot \text{weight}(\overline{\pi}, [\frac{1}{3n}, \frac{2}{n}])$.*

As an illustration, if $\pi$ is the Zipf distribution and the number of clusters $N \geq n$ then $\tau_1 = \Omega(1/n)$ and $\text{weight}(\overline{\pi}, [0, 1/n]) = \Omega(1)$ (see (Feldman, 2020) for more examples). On the contrary, when $\pi$ is not heavy-tailed, $\tau_1$ will be small.

**Lemma B.5** (Lemma 2.7 in (Feldman, 2020)). *Let $\pi$ be a frequency prior such that for some $\theta \leq 1/(2n)$, weight$(\bar{\pi}, [\theta, t/n]) = 0$, where $t = \ln(1/(\theta\beta))$, $\beta = $ weight$(\bar{\pi}, [0, \theta])$. Then $\tau_1 \leq 2\theta$.*

The above lemma indicates that when the frequencies are not heavy-tailed then $\tau_1$ is small (and hence generalization is not hurt by not memorizing).

### B.4. The Effects of Noise

In this section we analyze the effect of adding noise to the training set. We distinguish two cases: the low noise regime and the high noise regime.

**Low Noise Regime.** When the noise level is small, the originally separated subpopulations (at $t = 0$) will remain separated. This implies that if the frequencies of the subpopulations were originally heavy-tailed (as in the above discussion), they will remain heavy-tailed even in the low-noise regime. This will imply that some clusters will be represented by singletons ($\ell = 1$) and any algorithm that satisfies errn$_Z(A, 1) \neq 0$ has to pay $\tau_1 \cdot$ errn$_Z(A, 1)$ in the population error with $\tau_1$ being lower bounded as in Lemma B.4. We interpret errn$_Z(A, 1) \approx 0$ as evidence for memorization. To be more concrete, we will need the following definition that is a smooth generalization of single representative of a subpopulation.

**Definition B.6.** We will say that a subpopulation $C$ has an $\epsilon$-smoothed single representative in a set of points $S$ belonging to $C$ if for any $x, x' \in S$, it holds that $\|x - x'\| \leq \epsilon$.

Intuitively this means that if there are more than one images in the training set $Z$ from $C$, they are all very close to each other. This will be the case in diffusion with low noise.

**Lemma B.7** (Subpopulations Remain Heavy-Tailed). *Consider an example $x_0 \in \mathbb{R}^d$ that is the unique representative of a subpopulation $j \in [N]$ in the training set $Z$ with $\|x_0\| \leq$ poly$(d)$. Consider $m$ noisy copies $\{x_t^i\}_{i \in [m]}$ of $x_0$ at noise level $t : x_t^i = \sqrt{1 - \sigma_t^2} x_0 + \sigma_t z_t^i, z_t^i \sim \mathcal{N}(0, I_d)$. Then the subpopulation $j$ has a poly$(1/d)$-smoothed single representative in the set $\{x_t^i\}_{i \in [m]}$ for $\sigma_t = $ poly$(1/d)$ with probability at least $1 - m \exp(-d/2)$.*

The proof appears in Section D.2.2. The above lemma implies that if the original dataset contains various well separated images (in the sense that correspond to representatives of single subpopulations), then after adding noise to each one of them (and even if we create multiple copies for each example), the clusters will remain separated when $\sigma_t$ is small. This implies that the single subpopulations remain and Lemma B.4 applies ($\tau_1$ is large).

For an illustration, let us consider the GMM density function $q = \sum_{i=1}^N w_i \mathcal{N}(\mu_i, I)$. It is a standard calculation to see that at time $t$, the pdf of the forward diffusion process is $q_t = \sum_{i=1}^N w_i \mathcal{N}(\sqrt{1 - \sigma_t^2}\mu_i, I)$, which means that the clusters are starting to concentrate around 0 as $t \to 1$ and the images from different subpopulations are starting to look more and more indistinguishable (since the TV distance between the components is contracting with $t$). We will say that two components $\mathcal{N}, \mathcal{N}'$ are $\epsilon$-separated if TV$(\mathcal{N}, \mathcal{N}') > 2\epsilon$.

**Lemma B.8** (Clusters Are Separated in Low Noise). *Any pair of Gaussians with original total variation $C = 1/600$ will be $\epsilon$-separated at noise scale $\sigma_t \leq \sqrt{1 - (2\epsilon/C)^2}$.*

For the proof, see Section D.2.3.

**High Noise Regime.** As we increase $t$, we add more and more noise to the images. This means that the clusters start to merge and the heavy-tailed distribution of the mixing coefficients becomes lighter (until all the clusters are merged into a single one). To illustrate this phenomenon, we will consider a mixture of Gaussians, which is the standard model for clustering tasks (we expect similar behavior for more general mixture models). Let us again consider the density function $q = \sum_{i=1}^N w_i \mathcal{N}(\mu_i, I)$. Also, let the pdf of the forward diffusion process be $q_t = \sum_{i=1}^N w_i \mathcal{N}(\sqrt{1 - \sigma_t^2}\mu_i, I)$. We will say that two components $\mathcal{N}, \mathcal{N}'$ can be $\epsilon$-merged if TV$(\mathcal{N}, \mathcal{N}') \leq \epsilon$.

**Lemma B.9** (Clusters Merge in High Noise). *Any pair of Gaussians with original total variation $C = 1/600$ will be $\epsilon$-merged at noise scale $\sigma_t \geq \sqrt{1 - (\epsilon/C)^2}$.*

For the proof, see Section D.2.3. As the clusters are getting merged, then their coefficients are added up and their distribution is no more heavy-tailed. Hence, Lemma B.5 implies that $\tau_1$ will be small. This conceptually indicates that there is no reason for memorizing the training noisy images $x_t$ (and hence the original images $x_0$ which do not appear during training).

Given the above discussion, we reach the conclusion that if the frequencies of the original subpopulations are heavy-tailed then, in the low-noise regime, the training set will have single subpopulations and, in that case, fitting these single

representatives is required for successful generalization. However, in the high-noise regime, the noisy training set does not have isolated examples and, in principle, there is no reason to memorize its elements (and hence even elements of the original set). We believe that this discussion sheds some light on the nature of memorization needed for optimal generative modeling and motivates our training Algorithm 1 that avoids memorization only in the high-noise regime.

## C. Noisy data training of stable diffusion using v-prediction

The variance-preserving forward process defines the following transition probability distribution:

$$p(X_t = x_t | X_0) = \mathcal{N}(x_t; \alpha_t X_0, \sigma_t^2 I) \text{ and } p(X_t = x_t | X_s) = \mathcal{N}(x_t; (\alpha_t/\alpha_s) X_s, \sigma_{t|s}^2 I).$$

where $\sigma_{t|s}^2 = (1 - \frac{\alpha_t^2 \sigma_s^2}{\sigma_t^2 \alpha_s^2})\sigma_t^2$. Let $t_{\mathrm{n}}$ be the noise scale corresponding to the noisy data and the noisy data $x_{t_{\mathrm{n}}}$ from the clean data $x_0$ has the probability distribution $p(X_{t_{\mathrm{n}}} = x_{t_{\mathrm{n}}} | X_0) = \mathcal{N}(x_{t_{\mathrm{n}}}; \alpha_{t_{\mathrm{n}}} X_0, \sigma_{t_{\mathrm{n}}}^2 I)$. In this case, the following Lemma holds.

**Lemma C.1.** $\mathbb{E}[X_{t_{\mathrm{n}}} | X_t] = \frac{\alpha_{t_{\mathrm{n}}} \sigma_{t|t_{\mathrm{n}}}^2}{\sigma_t^2} \mathbb{E}[X_0 | X_t] + \frac{\alpha_t \sigma_{t_{\mathrm{n}}}^2}{\sigma_t^2 \alpha_{t_{\mathrm{n}}}} X_t.$

*Proof.* Let $p_t(\cdot)$ denote the probability density of the random variable $X_t$. Observe that $X_t = \alpha_t X_0 + \sigma_t Z$. Using Tweedie's formula, we have

$$\nabla \log p_t(X_t) = \frac{\alpha_t \mathbb{E}[X_0 | X_t] - X_t}{\sigma_t^2}.$$

Additionally, the random variable $X_t = (\alpha_t/\alpha_{t_{\mathrm{n}}}) X_{t_{\mathrm{n}}} + \sigma_{t|t_{\mathrm{n}}} Z$. Using Tweedie's formula, we can write the score function

$$\nabla \log p_t(X_t) = \frac{(\alpha_t/\alpha_{t_{\mathrm{n}}}) \mathbb{E}[X_{t_{\mathrm{n}}} | X_t] - X_t}{\sigma_{t|t_{\mathrm{n}}}^2}.$$

Using the above two equations, we have

$$\frac{(\alpha_t/\alpha_{t_{\mathrm{n}}}) \mathbb{E}[X_{t_{\mathrm{n}}} | X_t] - X_t}{\sigma_{t|t_{\mathrm{n}}}^2} = \frac{\alpha_t \mathbb{E}[X_0 | X_t] - X_t}{\sigma_t^2}$$

$$\mathbb{E}[X_{t_{\mathrm{n}}} | X_t] = \frac{\alpha_{t_{\mathrm{n}}} \sigma_{t|t_{\mathrm{n}}}^2}{\alpha_t \sigma_t^2} (\alpha_t \mathbb{E}[X_0 | X_t] - X_t) + \frac{\alpha_{t_{\mathrm{n}}} X_t}{\alpha_t} = \frac{\alpha_{t_{\mathrm{n}}} \sigma_{t|t_{\mathrm{n}}}^2}{\sigma_t^2} \mathbb{E}[X_0 | X_t] + \frac{\alpha_t \sigma_{t_{\mathrm{n}}}^2}{\sigma_t^2 \alpha_{t_{\mathrm{n}}}} X_t$$

.

$\square$

**Lemma C.2.** *Predicting* $\alpha_t Z - \sigma_t \frac{(X_{t_{\mathrm{n}}} - \frac{\alpha_t \sigma_{t_{\mathrm{n}}}^2}{\sigma_t^2 \alpha_{t_{\mathrm{n}}}} X_t)}{\frac{\alpha_{t_{\mathrm{n}}} \sigma_{t|t_{\mathrm{n}}}^2}{\sigma_t^2}}$ *gives us that the optimal v-prediction.*

## D. Proofs

### D.1. Technical Details about Information Leakage

#### D.1.1. PROOF OF LEMMA 4.1

*Proof.* The distribution of the training data conditioned on the dataset $S_{t_{\mathrm{n}}}$ is $q_0(x) = \frac{1}{n} \sum_{x_{t_{\mathrm{n}}} \in S_{t_{\mathrm{n}}}} \delta(x - x_{t_{\mathrm{n}}})$. To obtain iterates at time $t$, we add additional noise to points $x_{t_{\mathrm{n}}} \in S_{t_{\mathrm{n}}}$. Particularly, the following relation holds for any $t \in (t_{\mathrm{n}}, T]$:

$$X_t^{\mathrm{Amb}} = \sqrt{\frac{1 - \sigma_t^2}{1 - \sigma_{t_{\mathrm{n}}}^2}} X_{t_{\mathrm{n}}} + \sqrt{\frac{\sigma_t^2 - \sigma_{t_{\mathrm{n}}}^2}{1 - \sigma_{t_{\mathrm{n}}}^2}} \epsilon, \epsilon \sim \mathcal{N}(0, I).$$

This induces a distribution for each time $t$:

$$q_t(x | S_{t_{\mathrm{n}}}) = \frac{1}{n} \sum_{x_{t_{\mathrm{n}}} \in S_{t_{\mathrm{n}}}} \mathcal{N}\left(x; \sqrt{\frac{1 - \sigma_t^2}{1 - \sigma_{t_{\mathrm{n}}}^2}} x_{t_{\mathrm{n}}}, \frac{\sigma_t^2 - \sigma_{t_{\mathrm{n}}}^2}{1 - \sigma_{t_{\mathrm{n}}}^2} \cdot I\right).$$

The score of the Gaussian mixture $q_t$ is given by

$$s_t^{\text{Amb}}(x|S_{t_{\text{n}}}) = \frac{1}{\frac{\sigma_t^2 - \sigma_{t_{\text{n}}}^2}{1 - \sigma_{t_{\text{n}}}^2}} \sum_{x_{t_{\text{n}}} \in S_{t_{\text{n}}}} \left( \sqrt{\frac{1 - \sigma_t^2}{1 - \sigma_{t_{\text{n}}}^2}} x_{t_{\text{n}}} - x \right) \frac{\mathcal{N}\left(x; \sqrt{\frac{1 - \sigma_t^2}{1 - \sigma_{t_{\text{n}}}^2}} x_{t_{\text{n}}}, \frac{\sigma_t^2 - \sigma_{t_{\text{n}}}^2}{1 - \sigma_{t_{\text{n}}}^2} \cdot I\right)}{\sum_{y \in S_{t_{\text{n}}}} \mathcal{N}\left(x; \sqrt{\frac{1 - \sigma_t^2}{1 - \sigma_{t_{\text{n}}}^2}} y, \frac{\sigma_t^2 - \sigma_{t_{\text{n}}}^2}{1 - \sigma_{t_{\text{n}}}^2} \cdot I\right)} .$$

Since the reverse flow of Eq.(5) provably reverses the forward diffusion (Song et al., 2020), the distribution $q_0^{\leftarrow}$ equals the empirical data distribution $q_0$, which is a sum of delta functions on the noisy training set $S_{t_{\text{n}}}$. $\qquad\square$

### D.1.2. PROOF OF LEMMA 4.2

*Proof.* For two random variables $X, Y$, recall that $I(X; Y) = H(X) + H(Y) - H(X, Y)$, where $H(X)$ is the entropy of $X$ and $H(X, Y)$ is the joint entropy of $X$ and $Y$. Without loss of generality, let $\mu = 0$. Let $x_0 \sim \mathcal{N}(0, \Sigma)$. For the Ambient Diffusion at time $t_{\text{n}}$, conditional on the noisy point being $x_{t_{\text{n}}}$, the optimal distribution learned is $\delta(x - x_{t_{\text{n}}})$, where $x_{t_{\text{n}}} = \sqrt{1 - \sigma_{t_{\text{n}}}^2} x_0 + \sigma_{t_{\text{n}}} Z$. Note that $n$ i.i.d. draws from this distribution (denoted by $A$) are identical and hence

$$I(A; x_0) = I(x_{t_{\text{n}}}; x_0) .$$

Now observe that

$$x_0 \sim \mathcal{N}(0, \Sigma)$$

and

$$x_{t_{\text{n}}} \sim \mathcal{N}(0, (1 - \sigma_{t_{\text{n}}}^2)\Sigma + \sigma_{t_{\text{n}}}^2 I) .$$

Moreover, for the random column vector $\zeta = [x_0, x_{t_{\text{n}}}]^\top$, we have that $\mathbb{E}[\zeta\zeta^\top] = \left[ \begin{array}{c|c} \Sigma & \sqrt{1 - \sigma_{t_{\text{n}}}^2} \Sigma \\ \hline \sqrt{1 - \sigma_{t_{\text{n}}}^2} \Sigma & (1 - \sigma_{t_{\text{n}}}^2)\Sigma + \sigma_{t_{\text{n}}}^2 I \end{array} \right]$.

Now it remains to control the mutual information of Gaussians. Given that $\Sigma^{-1}$ exists, we note that $\det(\mathbb{E}[\zeta\zeta^\top]) = \det(\Sigma) \cdot \det(\sigma_{t_{\text{n}}}^2 I)$. We can hence write

$$I(x_{t_{\text{n}}}; x_0) = \frac{1}{2} \log \frac{\det(\mathbb{E}[x_0 x_0^\top])\det(\mathbb{E}[x_{t_{\text{n}}} x_{t_{\text{n}}}^\top])}{\det(\mathbb{E}[\zeta\zeta^\top])} = \frac{1}{2} \log \frac{\det((1 - \sigma_{t_{\text{n}}}^2)\Sigma + \sigma_{t_{\text{n}}}^2 I)}{\det(\sigma_{t_{\text{n}}}^2 I)} . \tag{11}$$

This simplifies to

$$\frac{1}{2} \log \det \left( I + \frac{1 - \sigma_{t_{\text{n}}}^2}{\sigma_{t_{\text{n}}}^2} \Sigma \right) ,$$

where $\frac{1 - \sigma_{t_{\text{n}}}^2}{\sigma_{t_{\text{n}}}^2}$ corresponds to the signal-to-noise ratio. (An equivalent way to see the above, is by taking the conditional distribution $x_{t_{\text{n}}}|x_0$, which has covariance $\sigma_{t_{\text{n}}}^2 I$, and hence directly get (11).)

On the other side, for DDPM, the learned distribution is $\mathcal{N}(\sqrt{1 - \sigma_{t_{\text{n}}}^2} x_0, \sigma_{t_{\text{n}}}^2 I)$. Let $X_1$ be a single draw from that distribution. Hence, $m$ i.i.d. draws $S$ from that measure correspond to mutual information

$$I(S; x_0) = m \cdot I(X_1; x_0) = m \cdot I(x_{t_{\text{n}}}; x_0) .$$

This concludes the proof. $\qquad\square$

### D.1.3. ADDITIONAL BOUNDS ON MUTUAL INFORMATION FOR AMBIENT DIFFUSION

The following lemma gives a bound on the mutual information of a generated set of size $m$ from Ambient Diffusion at time $t_{\text{n}}$ given a training set of size $N$. On the other side, the mutual information of the DDPM solution at time t-nature should be $m$ times larger.

**Lemma D.1.** *Consider a dataset $S$ of size $N$ drawn i.i.d. from $\mathcal{N}(\mu, I)$. Consider the optimal ambient solution at time $t_{\text{n}}$ with input $S$. Consider a set $A$ of size $m$ generated i.i.d. by that distribution. Then $I(D; S) \leq md/2 \cdot \log(1/\sigma_{t_{\text{n}}}^2)$.*

*Proof.* Let $X_i$ be $N$ i.i.d. draws from $\mathcal{N}(\mu, I)$. For each $i$, let $Y_i = (1 - \sigma_{t_n}^2)X_i + \sigma_{t_n}Z_i$ for some independent normal $Z_i \sim \mathcal{N}(0, I)$. We know that the optimal ambient solution is the empirical distribution

$$P(y) = \frac{1}{N}\sum_i \delta(y - Y_i),$$

conditioned on the realization of the noisy dataset $Y = \{Y_1, ..., Y_N\}$.

By the data processing inequality, we know that $I(A; S) \leq I(Y; S)$. Since $Y_i$ are generated all in the same way and independently, we can write

$$I(Y; S) = \sum_i I(Y_i; X_i) = mI(Y_1; X_1).$$

We have that $I(Y_1; X_1) = H(Y_1) - H(Y_1|X_1)$. Recall that $X_1 \sim \mathcal{N}(\mu, I)$ and $Y_1|X_1 \sim \mathcal{N}((1 - \sigma_{t_n}^2)X_1, \sigma_{t_n}^2 I)$. Moreover, note that $Y_1 \sim \mathcal{N}((1 - \sigma_{t_n}^2)\mu, I)$. These imply that

$$H(Y_1) = \frac{d}{2}\log(2\pi e)$$

(since it has identity covariance) and

$$H(Y_1|X_1) = \frac{d}{2}\log(2\pi e\sigma_{t_n}^2).$$

This means that

$$I(Y_1; X_1) = \frac{d}{2}\log(1/\sigma_{t_n}^2).$$

This concludes the proof. $\qquad\square$

### D.2. Technical Details about the Subpopulations Model

#### D.2.1. PROOF THEOREM B.2

*Proof.* For each subpopulation with exactly $\ell$ representatives, we put in the set $X_{Z\#\ell}$ those representatives. Observe that the collection of sets $\{X_{Z\#\ell}\}$ partitions $Z$ for $\ell \in \{1, ..., n\}$. Set $X_Z = \cup_{\ell \in [n]} X_{Z\#\ell}$. The unseen points correspond to the set $X_{Z\#0}$.

With this notation in hand, we define

$$\mathrm{errn}_Z(A, \ell) = \mathbb{E}_{s_\theta \sim A(Z)} \sum_{x \in X_{Z\#\ell}} M_{i_x}(x)L(s_\theta, x).$$

where $i_x$ is the index of the unique component whose support contains $x$. We have that

$$\overline{\mathrm{err}}(\pi, A|Z) = \mathbb{E}_{D \sim D_\pi^X(\cdot|Z)}\mathbb{E}_{s_\theta \sim A(Z)} \sum_{x \in X} M_D(x) \cdot L(s_\theta, x).$$

We now decompose $X = X_Z \cup X_{Z\#0}$ and write

$$\overline{\mathrm{err}}(\pi, A|Z) = \sum_{x \in X_Z} \mathbb{E}_{D, s_\theta}[M_D(x) \cdot L(s_\theta, x)] + \sum_{x \in X_{Z\#0}} \mathbb{E}_{D, s_\theta}[M_D(x) \cdot L(s_\theta, x)].$$

Let us first deal with the second term. For any $x \in X_{Z\#0}$, it holds

$$\mathbb{E}_{D, s_\theta}[M_D(x) \cdot L(s_\theta, x)] = \mathbb{E}_{D \sim D_\pi^X(\cdot|Z)}M_D(x) \cdot \mathbb{E}_{s_\theta \sim A(Z)}L(s_\theta, x),$$

because the way we choose $D$ is independent of the random variable $L(s_\theta, x)$ which only depends on the way the algorithm picks the score function given the dataset.

Set $p(x, Z) = \mathbb{E}_{D \sim D_\pi^X(\cdot|Z)} M_D(x)$. Hence, from the elements that do not appear in $Z$, we get a contribution

$$\sum_{x \in X_{Z \#0}} p(x, Z) \cdot \mathbb{E}_{s_\theta \sim A(Z)} L(s_\theta, x). \tag{12}$$

Let us now deal with the elements appearing in $Z$. Fix $\ell \in [n]$. For any $x \in X_{Z \# \ell}$, we have that

$$\mathbb{E}_{D, s_\theta}[M_D(x) \cdot L(s_\theta, x)] = \mathbb{E}_{D \sim D_\pi^X(\cdot|Z)}[M_D(x)] \cdot \mathbb{E}_{s_\theta \sim A(Z)} L(s_\theta, x),$$

since the random variables $L(s_\theta, x)$ and $M_D(x)$ are independent given $Z$. By Lemma 2.1 in (Feldman, 2020) and since the supports of the components $M_1, ..., M_N$ are disjoint, we know that $\mathbb{E}_{D \sim D_\pi^X(\cdot|Z)}[M_D(x)] = \mathbb{E}[D(i_x) M_{i_x}(x)] = \mathbb{E}[D(i_x)] M_{i_x}(x) = \tau_\ell M_{i_x}(x)$, where $i_x$ is the index of the component whose support contains $x$. Hence, we have that

$$\sum_{x \in X_Z} \mathbb{E}[M_D(x) \cdot L(s_\theta, x)] = \sum_{\ell \in [n]} \sum_{x \in X_{Z \# \ell}} \tau_\ell \cdot M_{i_x}(x) \cdot \mathbb{E}_{s_\theta \sim A(Z)} L(s_\theta, x) = \sum_\ell \tau_\ell \cdot \sum_{x \in X_{Z \# \ell}} M_{i_x}(x) \mathbb{E}_{s_\theta \sim A(Z)} L(s_\theta, x)$$

In total, we have shown that

$$\overline{\mathrm{err}}(\pi, A|Z) = \sum_{\ell \in [n]} \tau_\ell \cdot \mathrm{errn}_Z(A, \ell) + \overline{\mathrm{err}}_{\mathrm{unseen}}(\pi, A|Z).$$

This completes the proof. $\qquad\square$

### D.2.2. PROOF OF LEMMA B.7

*Proof of Lemma B.7.* We have that the $i$-th noisy example can be written as $x_t^i = \sqrt{1 - \sigma_t^2} x_0^i + \sigma_t z_t^i$. Let us set $\sigma_t = o(1/\|x_0\|) = \mathrm{poly}(1/d)$. Using Taylor's approximation for $\sqrt{1-x}$ around $x = 0$ ($\sqrt{1-x} = 1 - x/2 - o(x)$), we can write

$$\|x_t^i - (1 - \mathrm{poly}(1/d)) x_0 - \mathrm{poly}(1/d) z_t^i\| \leq \epsilon,$$

for some $\epsilon = \mathrm{poly}(1/d)$ sufficiently small. This means that

$$\|x_t^i - x_0\| \leq \epsilon + \mathrm{poly}(1/d)\|x_0\| + \mathrm{poly}(1/d)\|z_t^i\|$$

By Gaussian concentration, we have that

$$\Pr_{z_t^i}[\|z_t^i\| > \sqrt{d}] \leq \exp(-d/2).$$

Let us define the bad event $E_m$ which corresponds to "subpopulation $j$ does not have a poly$(1/d)$-smoothed single representative in the set $\{x_t^i\}_{i \in [m]}$ for $\sigma_t = \mathrm{poly}(1/d)$". A union bound over the $m$ noisy examples gives that

$$\Pr_{z_t^1, ..., z_t^m}[E_m] \leq m \cdot \exp(-d/2).$$

$\qquad\square$

### D.2.3. PROOFS OF LEMMA B.8 AND LEMMA B.9

*Proofs of Lemma B.8 and Lemma B.9.* The proof relies on the fact that when the total variation distance between two identity-covariance Gaussians is smaller than an absolute constant, then the total variation is up to constants characterized by the distance between the means (Arbas et al., 2023). When the original total variation is at most $1/600$, (Arbas et al., 2023) shows that

$$\mathrm{TV}(\mathcal{N}(\mu, I), \mathcal{N}(\mu', I)) = \Theta(\|\mu - \mu'\|).$$

The lemmas follow by noting that the densities of $\mathcal{N}(\mu_i, I), \mathcal{N}(\mu_i', I)$ at noise scale $\sigma_t$ (denoted as $\mathcal{N}_t, \mathcal{N}_t'$) satisfy

$$\mathcal{N}_t = \mathcal{N}(\sqrt{1 - \sigma_t^2}\mu, I), \quad \mathcal{N}_t' = \mathcal{N}(\sqrt{1 - \sigma_t^2}\mu', I).$$

Since $\sqrt{1 - \sigma_t^2} \leq 1$, the means are contracting and so $\mathrm{TV}(\mathcal{N}_t, \mathcal{N}_t') \leq 1/600$. Also:

$$\mathrm{TV}(\mathcal{N}_t, \mathcal{N}_t') \leq \|\mu_t - \mu_t'\|/\sqrt{2} = \sqrt{1 - \sigma_t^2} \cdot \|\mu - \mu'\|/\sqrt{2}\,.$$

If we want to make this quantity at most $\epsilon$, it suffices to take $\sigma_t \geq \sqrt{1 - 2(\epsilon/\|\mu - \mu'\|)^2}$.

For the other side, by (Arbas et al., 2023), $\mathrm{TV}(\mathcal{N}_t, \mathcal{N}_t') \geq \|\mu_t - \mu_t'\|/200 = \sqrt{1 - \sigma_t^2}\|\mu - \mu'\|/200$. (we assume that the original variation is smaller than $1/600$ and we contract it by adding noise). If this should be at least $2\epsilon$, then it should be that $\sigma_t \leq \sqrt{1 - 200^2(2\epsilon/\|\mu - \mu'\|)^2}$. This concludes the proof. $\qquad\square$

## E. Experimental Details.

For all of our experiments regarding unconditional generation, we use the Adam optimizer with a learning rate of 0.0001, betas (0.9, 0.999), an epsilon value of 1e-8, and a weight decay of 0.01. The model for FFHQ and CIFAR-10 is trained for 30,000 iterations with a batch size of 256 and the model for Imagenet is trained for 512 batch size for 80,000 iterations. For experiments on the Imagenet dataset, we train a class-conditional model.

For FFHQ and CIFAR-10 experiments, we randomly sample 300, 1000 and 3000 samples from the complete dataset to create the dataset with limited size. We use Tiny Imagenet dataset which consists of 200 classes (Le & Yang, 2015). We sample 5 images randomly from each class to create a dataset consisting of 1000. Similarly, we sample 15 images from each class to create a dataset consisting of 3000 images. For the unconditional and conditional generation experiments, we used with the implementation of (Karras et al., 2022) and default parameters of the implementation.

For text-conditioned experiments, we use the implementation of (Somepalli et al., 2023) and implement additional baseline (Wen et al., 2024) and our method in the implementation. Similar to previous works, we use LAION-10k dataset to train the stable diffusion v2 model for 100000 number of iterations using batch size 16. We use the final checkpoint after the complete training to evaluate the memorization, clipscore and fidelity. For the text-conditioned experiments, we tried adding nature noise at noise scale $\{25, 50, 100\}$ and chose the model with best image quality.

## F. Images generated using our method

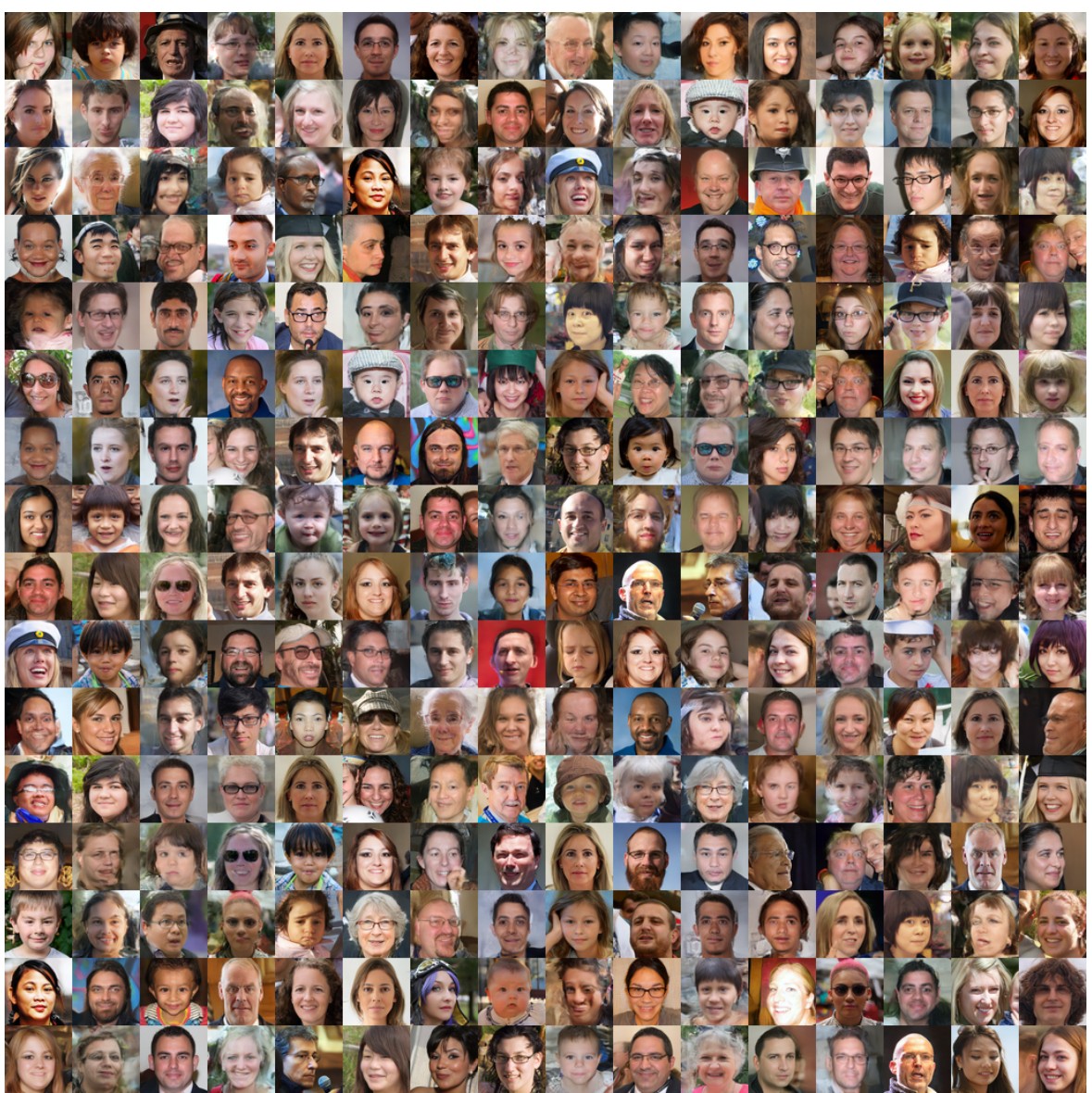

*Figure 4.* Images generated using a model trained with our method on 300 samples

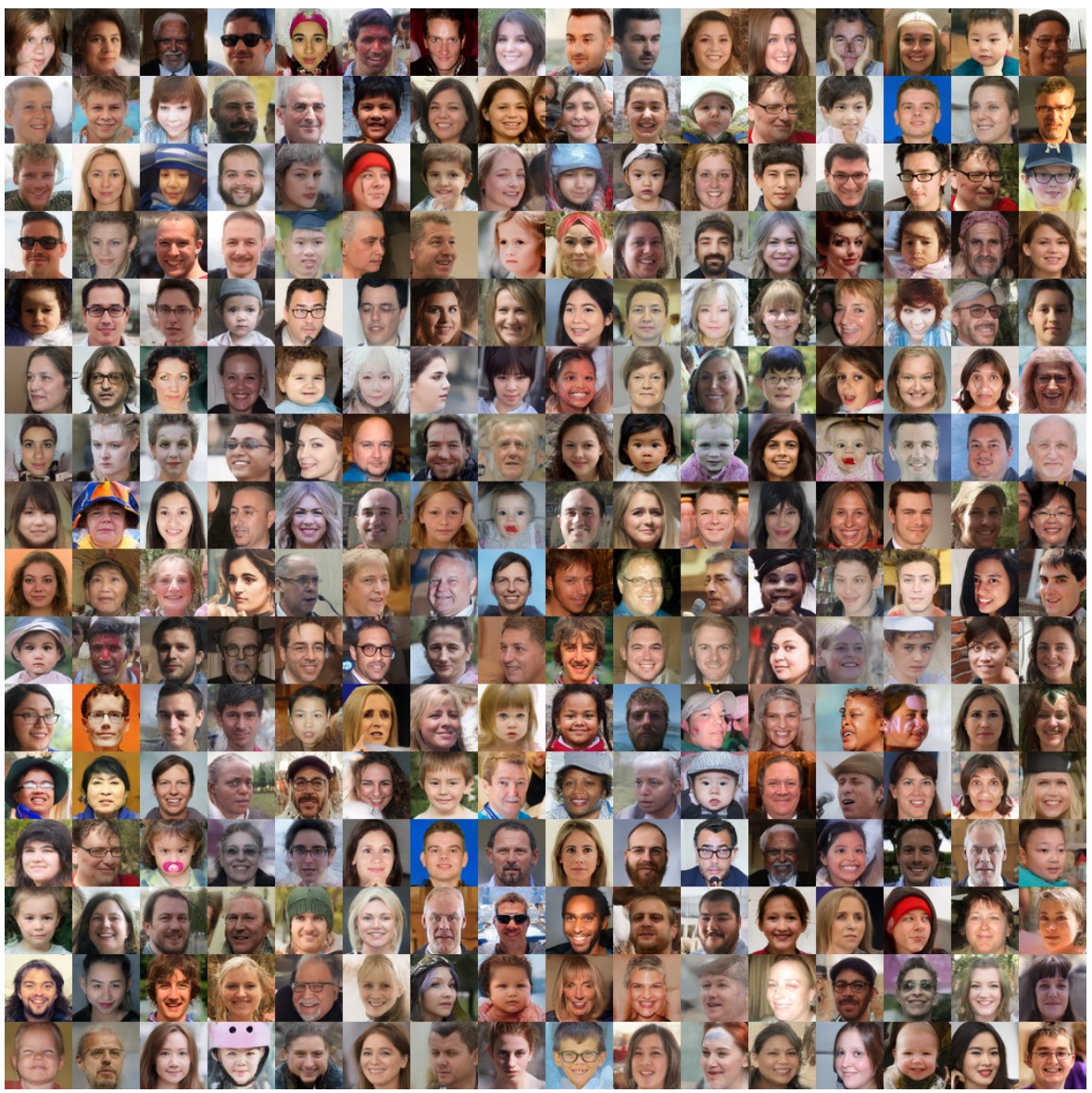

*Figure 5.* Images generated using a model trained with our method on 1000 samples

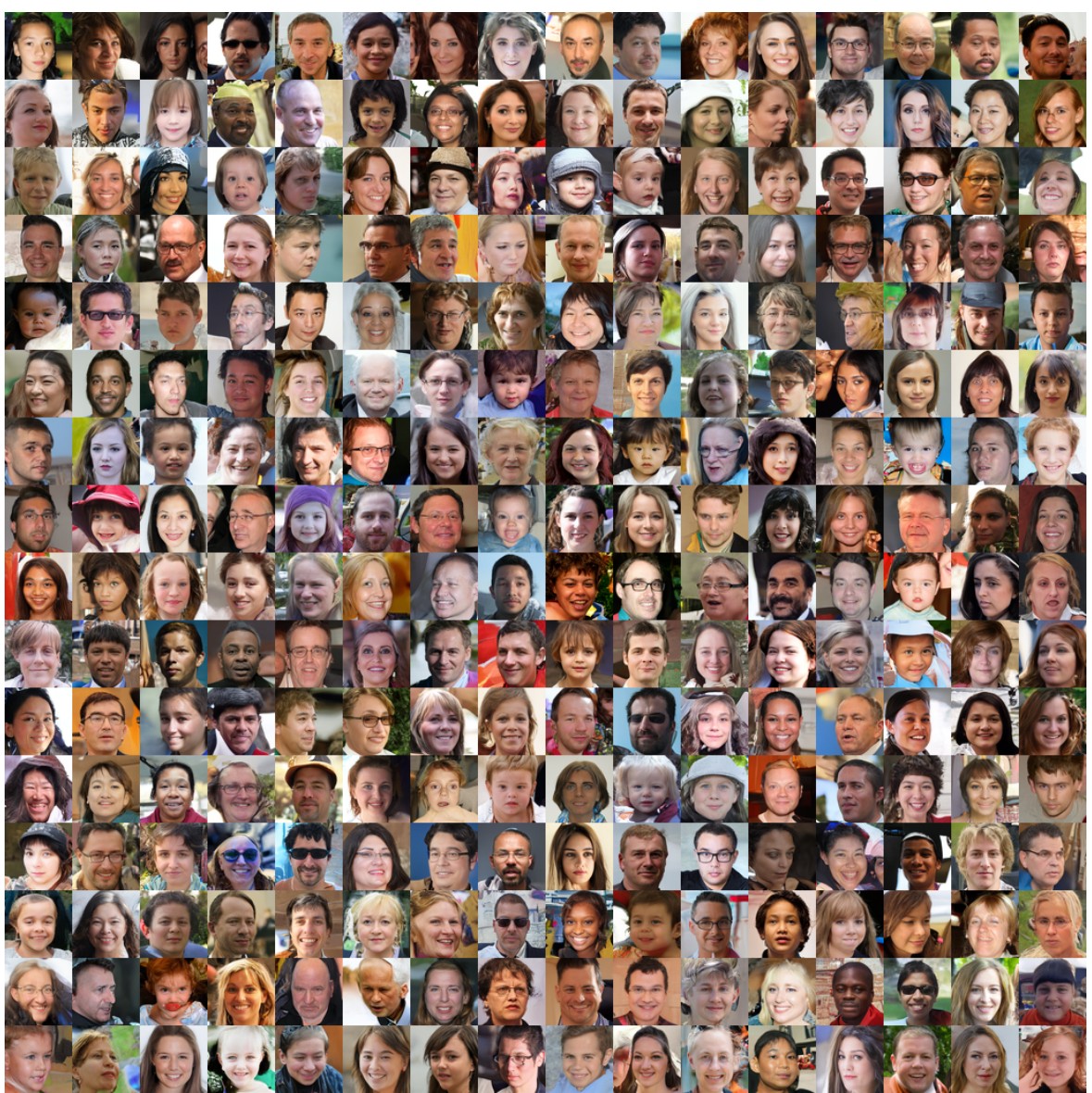

*Figure 6.* Images generated using a model trained with our method on 3000 samples

