# OpenReview forum: "Does Generation Require Memorization? Creative Diffusion Models using Ambient Diffusion"
_ICML.cc/2025/Conference — ICML 2025 poster_

### Official Review · Reviewer_SVEz · 2025-03-10

**Overall Recommendation:** 3

**Summary:**

This paper explores the use of noisy images to mitigate memorization issues in diffusion models, examining the tradeoff between fidelity and memorization. Under the assumption of normality in the data distribution, the authors analyze the problem of information leakage. They also investigate the memorization issue within a framework similar to that proposed by Feldman (2020). Empirical studies are conducted to better understand the fidelity–memorization tradeoff by introducing noise at different scales to the samples.

**Claims And Evidence:**

All the theoretical results are accompanied by proofs. And the empirical studies' conclusions are aligned with the provided empirical results.

**Essential References Not Discussed:**

It would be beneficial to include some work discussing the use of differential privacy-based methods to train generative models.

**Experimental Designs Or Analyses:**

The results in Table 1 are not well explained. For example, what is S. And for S>0.9, are they percentages of samples above the threshold? How are they computed? In addition, you mentioned that you played with t_n to obtain comparable FIDs of DDPM and yours, but the true values of t_n are not provided. In addition, could you also include more details on how the models are trained? E.g., have you used any clean images, etc. Currently, the relationship between FID and memorization issues and the selections of t_n remains unclear. Also, the Pareto frontier is not defined in the paper. Overall, it seems to me that the empirical studies are not comprehensive enough to give readers a good understanding of what the authors tried to present. Maybe more raw data and discussions on them should be made.

**Methods And Evaluation Criteria:**

Yes.

**Other Comments Or Suggestions:**

1. Eq (5) has the leading term -x. It seems like some discounting factor is missing.
2. The appearance of the formula at the bottom right of page 3 should be optimized.
3. Alg 1 should be included in the main text. (or a concise version should be included at least.)
4. Section 4.2 could be reorganized. Currently, readers might find it challenging to understand the main purpose of the section, even if a section overview has been provided.
5. Table 1 and Table 2 require more clarification.

**Other Strengths And Weaknesses:**

Strengths: Solving the memorization problems of the generative models is crucial. Solving this problem can potentially address copyright infringement problem, data providers' privacy concerns, etc.


Weakness:
1. I do not think the paper is well-organized and gets its theoretical results hinged with its empirical studies. The authors should have provided more intuitive descriptions of their theoretical work and how they together justify training diffusion models using generative models to tackle the memorization problems. In the current version, this kind of discussion is lacking, and it is difficult for the readers to see how the theoretical results are connected and how the empirical results support the theoretical claims.

2. Regarding the theoretical work, Lem 4.2 assumes that x0 ~ N(mu, sigma), which might be too strong. While I understand that the proof might be very challenging for a general data distribution, the authors should discuss why the current selection is sufficient to give a good understanding of the general case.

3. For the results in section 4.2, the assumed settings are not justified. In particular, I do not see the benefits of a mix of N distributions in understanding the memorization problem of the diffusion models. Besides, the derived results are not tailored for the diffusion model (i.e. the losses); as a result, it is somewhat questionable to what extent the theoretical results could characterize the diffusion models' memorization problem.

**Questions For Authors:**

Please the Other Strengths And Weaknesses section.

**Relation To Broader Scientific Literature:**

While the model is trying to address the memorization problem of the diffusion models, the work can be framed as using corrupted images to train a diffusion model to estimate the ground truth data distributions. This problem can be seen as a special case of the more general deconvolution theory in statistics.

Although not discussed in this paper, the memorization problem can also be tackled through the differential privacy-based method.

**Theoretical Claims:**

I checked those in section 4.1, which seem right. I am not familiar with the framework proposed by Feldman in section 4.2.1 but the results seem to be aligned with the intuition.

---

> ### Author Rebuttal · Authors · 2025-04-01
>
> From the questions raised, it appears that there are some major misunderstandings. In what follows, we do our best to clarify them, and we urge the reviewer to reread some part of our work and reassess their evaluation. In our end, we will do our best to improve the presentation based on the Reviewer's feedback.
>
> **"Have you used any clean images?"**: Yes, we use clean data but only for times $t\leq t_n$ of the diffusion, where $t_n$ is a hyperparameter to be controlled. For times $t\geq t_n$, we only use corrupted data and the Ambient Score Matching objective. The corruption to time $t_n$ happens once, before training, to ensure that there is no information leakage about the clean images at these times.
>
> **"what is S?"**: The meaning of S is explained on page 7, Lines 377-382 in the first column. To reiterate, for each generated image, we measure its similarity, S, (inner product), with the nearest neighbor in the dataset. Tables 1 (and Table 2) report percentages of the dataset that have similarity S greater than certain thresholds. In our rebuttal, we further report mean and median values -- see response to Reviewer Einp.
>
> **"the relationship between FID and memorization issues and the selections of $t_n$ remains unclear"**. Figure 1 shows the achieved FID and memorization for different values of $t_n$. There is an extended discussion on the choice of $t_n$ on page 7 of our submission.
>
> **"but the true values of t_n are not provided"**: this is a critical misunderstanding. We have access to clean data, and we intentionally corrupt it for some diffusion times to avoid memorization. Please refer to Algorithm 1 and Section 3 (Method, page 4) of our submission, where we verbally explain this algorithm. See also Choices for $t_n$ on page 7.
>
> **About differential privacy-based methods**: This is a great remark. We will add this discussion. Briefly, the notion of privacy is stronger than the notion of memorization. Privacy guarantees would mean that any adversary that has access to the model could not extract samples used for the training of that model. Our method does not guarantee that: we show that sampling with our model leads to outputs that resemble less to the training points. That does not exclude the possibility that an adversary with access to our model could recover the training points. Since we have relaxed expectations/guarantees, we also enjoy high performance.
>
> **On the Gaussian assumption in Lemma 4.2**: The goal of Lemma 4.2 is to motivate why the ambient optimal score solution ``leaks” less information at time $t_n$ compared to running vanilla DDPM from time $T$ until time $t_n$. While we believe that it holds for more general distributions, generalizing the result comes with some technical challenges and does not add much additional value to our goal of motivating why the ambient optimal score solution leaks less information.
>
> **Section 4.2**: Building a theoretical framework for understanding memorization in machine learning is quite challenging, and there is limited amount of work trying to address it. The seminal work of Feldman provides a natural instance (with some arguable caveats) of the learning problem where optimal classification requires memorization.
>
> Our work aims to take a step in understanding memorization in diffusion models, not by solely using Feldman’s framework but by taking a two-step approach:
>
> First, we adapt the setting of Feldman (which is a well-accepted) to more general loss functions (capturing e.g., standard score matching losses). The message of this general framework is roughly speaking the following: if the distribution of the frequencies of the subpopulations is heavy-tailed, then to achieve optimal generalization loss, the algorithm has to make the loss at all the points seen exactly once zero; otherwise, it pays a penalty \tau_1 for any not fitted point.
>
> Second, to make use of this general framework, we rely on how diffusion models are trained. We view the above general framework in many noise scales. As we explain e.g., in Lemma 4.5, even if the original dataset has the heavy-tailed structure of Feldman’s model, as the noise scale increases (which is the core idea of diffusion-based generation), the tails will become lighter since the populations start to merge. This is where our framework is specialized in diffusion models and not in Theorem 4.3.
>
> Combining the above two observations, the theoretical part of Section 4.2 gives evidence that avoiding memorization in the high-noise regime is feasible. Our main contribution in this part is the general observation that the tail of the distribution of the frequencies depends on the noise scale, and this is a property that depends crucially on how diffusion models are trained. This illustrates our conceptual contribution on why, for the high-noise regime, memorization could be avoided.
>
> We hope this rebuttal clarifies things and we remain at the Reviewer's availability if there are further concerns.

---

> > ### Comment · Reviewer_SVEz · 2025-04-03
> >
> > I thank the authors for the detailed clarifications. Most of my concerns are addressed. I have a few comments based on your responses:
> >
> > 1. Regarding "what is S?".  I believe this confusion was largely caused by some abuse of the notations. For example, S denotes a finite amount of examples in 2.2. While in Table 1, it becomes similarity and you also used S. Then in Table 2, it becomes Sim.  On page 7, Lines 377-382, I was aware of the definition, but the paper never introduced a dedicated notation for it. Regardless, this is a writing issue and I wish authors could make them consistent.
> >
> > 2. Regarding differential privacy, memorization and the discussion related. Thanks for the clarifications and it now makes sense. So the proposed method is only designed to alletivate the memorization problem instead of a protection of sensitive data (as the model still needs all clean versions of the dataset.) It will be helpful if you can put a note regarding this point as we have seen a sequence of work by Daras et al selling a similar idea from that perspective.
> >
> > 3. Regarding Lemma 4.2, as other reviewers also commented, while I could guess the motivation, the current way you present it is kind of misleading as the tone selling it appears like you assume a similar setting to Lemma 4.1. You may either consider extending it to the Lemma 4.1's setting or add addtional comments to better explain your motivation.
> >
> > 4. For the comments on Section 4.2, I like explanations, which make everything much clearer. It will be great if you can integrate them in the revision.
> >
> > I increased the score, although I still believe significant revisions to the presentation are needed.

---

> > > ### Author Response · Authors · 2025-04-08
> > >
> > > Thank you for raising your score and for your feedback!
> > >
> > > Regarding S, we acknowledge that the notation used could have been better. We will put a lot of effort into improving the presentation for the camera-ready to avoid confusing the reader. We will incorporate the discussion on memorization vs differential privacy, add additional comments for Lemma 4.2, and incorporate the discussion from the Rebuttal into Section 4.2.
> > >
> > > Thank you for your comments and we will make sure to significantly improve the clarify of the presentation for the camera-ready.

---

### Official Review · Reviewer_pCuc · 2025-03-11

**Overall Recommendation:** 3

**Summary:**

This paper addresses the issue of memorization in diffusion models, proposing a framework to reduce memorization while maintaining high-quality image generation. The authors introduce a simple method that utilizes noisy images to learn the initial portion of the generation trajectory, followed by high-quality images for the final portion. This approach effectively prevents memorization when forming the high-level structure of the generated image, while still allowing for high-quality, detailed features from the training images.
- The authors support their method with theoretical results that quantify the information leakage from the training set, demonstrating it to be smaller than standard approaches. They also extend existing results on memorization and generalization error from the classification literature to the context of diffusion models.
- They also empirically validate their approach, showing that it achieves a better tradeoff between memorization and quality compared to standard DDPM, and does a better job at preserving the quality of generated images compared to existing approaches for preventing memorization in diffusion models.

**Claims And Evidence:**

- The claims made in the submission are generally supported by clear evidence. The authors provide theoretical results, including Lemmas 4.1 and 4.2, which characterize the sampling distribution and compare mutual information between their method and standard diffusion. I have read the proofs of Lemmas 4.1, 4.2, and Theorem 4.3, and they seemed sound.
- The experimental results demonstrate the effectiveness of their approach in reducing memorization while maintaining image quality, as measured by FID scores and similarity metrics.
- While their theory does not completely speak to the success/guarantees of their particular method (for instance the information leakage guarantees hold only for ambient diffusion, and not their two-stage concatenated method), the authors are careful to point out these limitations.

**Essential References Not Discussed:**

I am not aware of any essential references that are missing.

**Experimental Designs Or Analyses:**

The experimental designs appear sound, thorough, and well-explained. The authors thoroughly investigate the memorization-quality tradeoff of their method along a number of axes, i.e. tuning the t_n parameter, varying dataset size, comparison to existing anti-memorization baselines, and extending their method to text-conditional generation.

**Methods And Evaluation Criteria:**

The methods and evaluation criteria appear appropriate and well-justified for the problem at hand. The authors use established metrics such as FID for image quality and nearest neighbor similarity scores for measuring memorization. The authors explore the effects of different t_n values and also compare against existing baselines for memorization prevention, in both cases looking at datasets of varying sizes, thus providing a thorough evaluation of the performance of their method.

**Other Comments Or Suggestions:**

- Several instances of straight quotes (") instead of proper LaTeX quotes (`` and '') appear throughout the paper, resulting in backwards-facing opening quotes.
- There are some grammatical issues that should be addressed, such as the phrase "How much heavy-tailed is the distribution" on lines 325-326.
- Lemma D.1 seems to have a typo as I assume the statement should be about I(A;S) rather than I(D;S).

**Other Strengths And Weaknesses:**

- For the most part, the paper is clearly written and easy to follow.
- I appreciate that their approach is simple, yet achieves convincing improvements wrt the tradeoff between memorization and quality.
- I found the appendix discussion of the theoretical results far easier to follow than the more informal discussion presented in the main portion. I understand that this may be due to a space constraint, but it would be worth taking a second look at the main portion to see how the discussion can be clarified.

**Questions For Authors:**

- As noted above, I would appreciate hearing from the authors in what ways the analysis presented in Section 4.2 is substantially different from [Feldman, 2020]

**Relation To Broader Scientific Literature:**

This work builds upon and extends existing literature on memorization in machine learning, particularly Feldman's work on heavy-tailed distributions. It also contributes to the growing body of research on diffusion models and their properties. It is notable that their algorithms produce state-of-art results in terms of the trade-off between memorization and quality for both image generation and text-conditional generation compared to prior work.

**Theoretical Claims:**

- The information leakage result (Lemmas 4.1/4.2) is a nice accompaniment for the proposed method, though I found it a bit unclear at first that the statement is about only a single point, and would have preferred that to be made more clear before the lemma was presented.
- The results in Section 4.2 make sense, but upon inspection of the proofs, it doesn't seem like they use anything specific to diffusion models. It would be useful for the authors to comment on if/how the proofs are substantially different from Feldman's results.
- In a similar vein, I found the statement of Theorem 4.3/B.2 to be a little hard to parse at first, especially when trying to compare to the analogous statement from Feldman, which compares the error to the optimal generalization error. Based on my understanding, the result implies this, but I think it might make it clear to add some discussion and/or changing how the theorem is presented.

---

> ### Author Rebuttal · Authors · 2025-04-01
>
> We thank the reviewer for careful reading, for their feedback and suggestions!
>
> **Lemma 4.2 being about a single point**: We preferred to present the statement for 1 point since it already presents the advantage of ambient diffusion compared to vanilla DDPM: Given a generation of m points from each of the two models, the mutual information is larger. We agree with the comment that a larger training set is desirable, and this is why we present such a result in the Appendix (we refer to it right after Lemma 4.2).
>
> **On the difference from Feldman’s work**: We believe that our main contribution in this part is the general observation that the tail of the distribution of the frequencies depends on the noise scale. This observation is specific to the way that diffusion models are trained and does not appear in Feldman’s work because it focuses on the multiclass classification. To make this observation more rigorous, we adapted Feldman’s framework with a more general loss function (which should be independent of the mixing coefficients): in fact, this adaptation is not technically challenging and we do not claim novel technical challenges; it is instead a generalization of the previous proof. However, the novelty comes from a conceptual perspective. As the noise level increases, the penalty \tau_1 in the Informal Theorem 1 is decreased because the tails become lighter.
>
> To summarize, this part of the paper illustrates our conceptual contribution on why for the high-noise regime, memorization could be avoided.
>
> **On the readability of appendix and main body**: We thank the reviewer for that comment. Indeed, the lack of space made the presentation a bit dense. We will use the extra page in the Camera Ready to improve the presentation of the work.
>
> We hope that our response addresses any remaining concerns and that the Reviewer will support the acceptance of our submission. We remain at the Reviewer's availability in case there are additional concerns to be addressed.

---

### Official Review · Reviewer_Einp · 2025-03-13

**Overall Recommendation:** 3

**Summary:**

The paper investigates the links between good performance of generative models (i.e., low FID) and memorization of the training set. in particular, the paper investigates the Pareto front. of performance vs memorization. Based on a recent "ambient score matching loss" [1], the paper introduces a new diffusion technique with good performance and less memorization.


[1] DARAS, Giannis, DIMAKIS, Alexandros G., et DASKALAKIS, Constantinos. Consistent diffusion meets tweedie: Training exact ambient diffusion models with noisy data. arXiv preprint ICML 2024

**Claims And Evidence:**

Yes

**Essential References Not Discussed:**

NA

**Experimental Designs Or Analyses:**

In the experimental results, the dataset ranges from 300 to 3k, does the proposed method also provide improvement for larger datasets? I.e. with the full dataset?

**Methods And Evaluation Criteria:**

In figure 2, how exactly do you define the "number of duplicates"? In [2] they use the cosine similarities between the synthetic image and the closest one from the training set. Maybe the mean/median of this cosine similarity could be a better than "number of duplicates", as done in hte experiment section as I understand.
- Would it be possible to have the same kind of plot with the Dinov2 FID between the generated set and the test set?

[2] Kadkhodaie, Zahra, et al. "Generalization in diffusion models arises from geometry-adaptive harmonic representations." ICLR (2024)


- **I did not understand the proposed algorithm**. How does the sampling process work exactly? You have 2 different time scales depending on how advanced the denoising process is.

**Other Comments Or Suggestions:**

NA

**Other Strengths And Weaknesses:**

NA

**Questions For Authors:**

NA

**Relation To Broader Scientific Literature:**

Good

**Theoretical Claims:**

- IMO the theoretical results lacks formality: I was not able to follow Section 4. It might be useful to properly encapsulate in an environment and explain one highlighted result.

---

> ### Author Rebuttal · Authors · 2025-03-31
>
> We thank the reviewer for their valuable reviews and suggestions!
>
> **On no of duplicates in figure 2**: By the no of duplicates, we mean the percentage of generated samples whose similarity to their nearest neighbor in the training set is greater than 0.9. The nearest neighbor is found using the cosine similarity in the DINO space.
>
> As per the reviewer’s request, we also calculated the mean and median of the similarity in the same setting as Figure 1 (300 training samples from FFHQ). We will include these results in the updated version.
>
> | Method | Mean | Median |
> |------------------------|:-------:|:-------:|
> | Baseline | 0.8826 | 0.8931 |
> | Ours ($\sigma=0.1$) | 0.8854 | 0.8748 |
> | Ours ($\sigma=0.25$) | 0.8518 | 0.8491 |
> | Ours ($\sigma=0.3$) | 0.8473 | 0.8475 |
> | Ours ($\sigma=0.4$) | 0.8287 | 0.8292 |
> | Ours ($\sigma=0.5$) | 0.8060 | 0.8069 |
>
> **On FID using Dinov2**:  As per the reviewer’s suggestion, we performed additional experiments to evaluate the quality of our model with FID using Dinov2. We will add these results with a plot in the updated version.
>
> | Method | FID | FD_{DinoV2} |
> |------------------------|:-------:|:-------:|
> | Baseline | 16.21 | 353.19 |
> | Ours ($\sigma = 0.1$) | 14.94 | 347.21 |
> | Ours ($\sigma = 0.25$) | 15.05 | 344.60 |
> | Ours ($\sigma = 0.3$) | 16.14 | 358.23 |
> | Ours ($\sigma = 0.4$) | 19.55 | 371.81 |
> | Ours ($\sigma = 0.5$) | 23.73 | 382.03 |
>
> **On clarification about the sampling process**: Our method is a training-time mitigation strategy, and therefore, no modifications are made during the sampling. For $t\geq t_n$, we replace the original dataset with a noisy version of it where each datapoint has been replaced (*once*) with a noisy realization. This step corresponds to the creation of $S_{t_n}$ at step 1 of Algorithm 1 (Appendix page 12). The creation of $S_{t_n}$ is very important as it ensures the reduced information about the clean distribution for the learning at $t\geq t_n$ (see Lemma 4.2). This step leads to decreased memorization. For $t\leq t_n$, we train as usual. We will clarify this in the updated version.
>
> **On theoretical results**: We will do our best to improve the presentation of our results, using the one additional page for the Camera Ready. Here we provide an intuitive explanation of the results, and will include such a discussion in the updated version to improve the readability.
>
> Lemma 4.1 proves that the optimal score learned at time $t_n$ is a gradient field that points towards the noisy training set. This lemma is formal and can be seen as the analogue of the optimal score solution for standard DDPM in a noiseless dataset.
>
> Lemma 4.2 motivates why this solution ``leaks’’ less information at time $t_n$ compared to running vanilla DDPM from time $T$ until time $t_n$. This property is formalized using the mutual information. Additionally, we needed the Gaussianity assumption to make it formal since for the Gaussian, we can derive explicit expressions for mutual information.
>
> Section 4.2 explores connections between our work and the work of Feldman, which focuses on the multiclass classification. Theorem 4.3 presents the analogue of Feldman’s result with a more general loss function. However, the message is essentially the same: the more heavy-tailed the distribution of the frequencies, the higher the penalty of not memorizing (in our setting, memorization corresponds to making the score objective small). This is formally presented in Lemma 4.4, which comes from the work of Feldman and shows that the penalty term \tau_1 depends on the tail of the distribution of the mixing coefficients. Finally, we illustrate in Lemma 4.5 that the noise scale of the diffusion process affects the parameter \tau_1 since, as the noise increases, the subpopulations merge and the mixing coefficients add up (hence becoming lighter). To make this intuitive argument fully formal, we provide the result for the most standard mixture model, namely Gaussian mixture, and the result should extend to any mixture family since noise addition is a contracting operation.
>
> **On larger datasets**: For the unconditional image generation setting, memorization becomes negligible for larger datasets. Per Reviewer’s Request, we trained on a slightly bigger dataset. For FFHQ 5k samples, we get the following results:
>
> | Method | FID | S > 0.85 | S > 0.875 | S > 0.9 |
> |---------|:-----:|:---------:|:---------:|:---------:|
> | Baseline | 6.4 | 21.58 | 4.98 | 0.46 |
> | Ours | 6.4 | 20.08 | 4.53 | 0.42 |
>
> As seen, our method is still a little better in terms of memorization with the same FID as the Baseline, but the improvement is smaller compared to more data-limited settings.
>
> For larger datasets, memorization is still an issue if there is text-conditioning. Our results in Section 5.2. show improvements over the baselines for this regime as well.
>
> We hope our rebuttal clarifies the Reviewer's concerns and that the Reviewer will strongly support the acceptance of our work.

---

### Official Review · Reviewer_pjLr · 2025-03-15

**Overall Recommendation:** 3

**Summary:**

In this paper, the authors propose a simple but effective method for the diffusion model to generate creative images rather than memorizing the training data. Their method is motivated by previous work of (Feldman, 2020) about the generalization in classification problems, where they showed that the model tends to memorize when the distribution of the frequencies is heavy-tailed. So they conjecture that the high noise added to the images in the diffusion model training process would force different subpopulations to start to merge and the heavy tail to disappear. Based on this intuition, they show that the memorization in diffusion model is only necessary in the low-noise region, and they use noisy data to train the model in the large-noise region. Some theory and experiments are provided to support their claims.

**Claims And Evidence:**

Most of their claims are supported by evidence. But I found some claims confusing:
* In the title of Figure 3, why does the "blurry" output stand for generalization? Can we view this as a quality degradation? (The authors have addressed this on their rebuttal.)

**Essential References Not Discussed:**

No.

**Experimental Designs Or Analyses:**

Yes, I reviewed the algorithm and the experimental results.

**Methods And Evaluation Criteria:**

I think the proposed method and the evaluation make sense for investigating the memorization of the diffusion model. Specifically, they validate the method on both the unconditional generation task and text-conditioned generation. The CIFAR-10, FFHQ, (tiny) ImageNet, and LAION are used to test the memorization.

**Other Comments Or Suggestions:**

I think the authors should have more discussion on related work in mitigating the memorization of diffusion models. There have been many interesting works in mitigating memorization, and you should discuss them in your work, although you have different techniques with them.

Some works in understanding and/or mitigating memorization that I know:
* On the Interpolation Effect of Score Smoothing. https://arxiv.org/abs/2502.19499
* Unveiling and Mitigating Memorization in Text-to-image Diffusion Models through Cross Attention. https://arxiv.org/abs/2403.11052v1
* Taking a Big Step: Large Learning Rates in Denoising Score Matching Prevent Memorization. https://arxiv.org/abs/2502.03435

Besides, I also suggest that you compare your method with those baselines in your experiments to validate your superiority.

**Other Strengths And Weaknesses:**

Please refer to other parts.

**Questions For Authors:**

Can your method be integrated with other diffusion methods such as EDM and DDIM? If yes, I would strongly recommend the authors to also apply your method on those diffusion piplines and verify the performance gain.

**Relation To Broader Scientific Literature:**

The proposed method involves two parts:
* For t>tn, they train the diffusion model using the Ambient Score Matching loss (Daras et al., 2024b)
* For t<tn, they train the diffusion model using DDPM.
It seems that you combine two existing methods and apply them on different noisy scales. Could you reclaim your algorithm novelty?

**Theoretical Claims:**

No, I didn't carefully check the proof in the appendix. But their explanation in the main paper make sense and they provide some intuition to help me understand.

---

> ### Author Rebuttal · Authors · 2025-03-31
>
> We thank the reviewer for their Review and suggestions.
>
> **On the blurry output of the diffusion models**: Figure 3 contains a noisy image $x_t$ and the learned model’s approximation of $E[x_0 | x_t]$. Intuitively, this expectation represents an average over all possible clean images $x_0$ that could have produced the noisy image $x_t$. As a result, the predicted image is expected to appear blurry. This is validated by the second row (outputs of a diffusion model trained on 52k images): a model trained on the full dataset has blurry predictions, as it should. On the other hand, when a model memorizes the training data, it is not modeling correctly the conditional expectation – instead, it is overconfident, i.e., it believes that one of a few clean images could correspond to a given noisy input $x_t$. In this case, it outputs a sharper (cleaner) image, which reflects memorization. The figure shows that our algorithm (Row 4) matches the blurriness level of the oracle model (52k training images) even though our model is only trained with 300 images.
>
> **On combining two existing algorithm and novelty**: The novelty of the algorithm is not in the loss, but in the way we create the data we apply the loss to. Let us clarify further. For times $t\geq t_n$, we replace the original dataset with a noisy version of it where each datapoint has been replaced (*once*) with a noisy realization.
> This step corresponds to the creation of the set $S_{t_n}$ at step 1 of Algorithm 1 (Line 612, Appendix Page 12). The fact that we create $S_{t_n}$ once before the launch of the training (instead of recreating at each epoch) is very important as it ensures that there is reduced information about the clean distribution for the learning that happens at times $t\geq t_n$ (see also Lemma 4.2). This step in the algorithm leads to decreased memorization.
>
> Comparison with existing works:
> - In the papers “Consistent Diffusion Meets Tweedie” and “How Much is a Noisy Image Worth?”, the authors are dealing with corrupted datasets and they are using Ambient Score Matching loss to train with the corrupted data.
> - In DDPM, clean data is used for all times using the regular objective.
> - In this paper, we have clean data, but we intentionally corrupt them for some times to reduce memorization. At the same time, we do not want to reduce performance, so we still use the clean data for certain diffusion times. The idea of intentional corruption (and the way its done in Algorithm 1) so that we achieve both reduced memorization and good performance is the algorithmic novelty of this work.
>
>
> **On the related work and additional baselines based on those works**: We want to bring to the Reviewers’ attention that **two out of the three papers that we are being asked to compare against were released last month** (26 Feb'25, and 5 Feb'25). We submitted this work in January 2025, so we do not think it is fair to compare against these two baselines. Besides, **neither of these two works has experiments with images**. That said, we are happy to cite and discuss these works in the updated version, and we will do so.
>
> We thank the Reviewer for bringing the paper “Unveiling and Mitigating Memorization in Text-to-image Diffusion Models through Cross Attention” to our attention and will cite it in the updated version. We attempted to compare against this method, but the code for training-time memorization mitigation is not provided in the official repository. We even tried implementing the approach ourselves, but couldn’t replicate their results. We contacted two of the authors about the issue, but as of today we have not received a reply. If the code becomes available, we will gladly compare.
>
> **On integrating the method with other diffusion methods**: Our method can work with any training architecture, training loss type (e.g., x0-prediction, epsilon-prediction, flow-matching, etc), and sampling method (EDM sampler, DDIM, DDPM sampler, etc). All the results in the paper are obtained by using the EDM codebase for all these components (architecture, training loss, sampler).
>
> As per Reviewer’s request, we experimented with other pipelines, and we present the results for (FID, Memorization) below when the training is performed using 300 training images for the FFHQ data set.
>
> | Method | FID | Memorization (S > 0.9) | Memorization (S > 0.875) | Memorization (S > 0.85) |
> |-------------------------------|-------|:------------------------:|:-------------------------:|:-------------------------:|
> | Baseline(ddim) | 16.34 | 41.62 | 49.8 | 58.14 |
> | Ours (ddim) | 16.48 | 23.76 | 33.58 | 43.60 |
> | Baseline (iddpm) | 16.83 | 45.54 | 53.74 | 61.82 |
> | Ours (iddpm) | 16.46 | 26.94 | 37.70 | 47.94 |
> | Baseline (ddpm (sde sampler)) | 16.73 | 42.37 | 51.91 | 59.48 |
> | Ours (ddpm (sde sampler)) | 16.35 | 24.34 | 36.28 | 46.04 |
>
> In the light of these new experiments and clarifications, we hope that the Reviewer will support the acceptance of our submission.

---

> > ### Comment · Reviewer_pjLr · 2025-04-01
> >
> > Thanks for the detailed response.
> >
> > I thought the pictures in Figure 3 were the final generated outputs of the diffusion model, so I had that concern. Thanks for your clarification that the picture is the one-step output of the denoiser (E[x_0|x_t]), so the results make sense. The blurry output indicates that the denoiser is combining multiple faces, and it learns the real underlying distribution rather than just memorizing training images. Maybe it would be more clear if you added some detail (E[x_0|x_t]) to the title of Figure 3.
> >
> > And I look forward to more comparison with the related work in your paper if possible in the future.
> >
> > The authors have addressed my concerns, so I increase my score to 3.

---

> > > ### Author Response · Authors · 2025-04-08
> > >
> > > Thank you for raising your score and for your valuable comments! We will update the title of Figure 3 as requested, and we will compare with the baseline you mentioned as soon as the code becomes available.

---

### Decision · Program_Chairs · 2025-05-01

**Decision:**

Accept (poster)

**Comment:**

This paper tackles memorization in diffusion models by proposing a framework that preserves generation quality while mitigating overfitting. The authors introduce a simple strategy that uses noisy images to guide the early stages of generation and high-quality images for the final stages. This approach prevents memorization of high-level structures while still capturing fine-grained details from the training data.

Most reviewers find the work well presented and interesting, which studies an important problem of memorization in diffusion models.
The paper is well supported by both empirical and theoretical studies. Please incorporate the reviewers' feedback into the revision.